# KLC4 shapes axon arbors during development and mediates adult behavior

**Elizabeth M Haynes[1,2,3,4], Korri H Burnett[1,3], Jiaye He[4,5], Marcel W Jean-Pierre[1,3], Martin Jarzyna[1,3], Kevin W Eliceiri[2,4], Jan Huisken[1,4,6], Mary C Halloran[1,3]***

[1]Department of Integrative Biology, University of Wisconsin-Madison, Madison, United States; [2]Center for Quantitative Cell Imaging, University of Wisconsin-Madison, Madison, United States; [3]Department of Neuroscience, University of Wisconsin-Madison, Madison, United States; [4]Morgridge Institute for Research, Madison, United States; [5]National Innovation Center for Advanced Medical Devices, Shenzen, China; [6]Department of Biology and Psychology, Georg-August-University, Göttingen, Germany

***For correspondence:**
mchalloran@wisc.edu

**Abstract** Development of elaborate and polarized neuronal morphology requires precisely regulated transport of cellular cargos by motor proteins such as kinesin-1. Kinesin-1 has numerous cellular cargos which must be delivered to unique neuronal compartments. The process by which this motor selectively transports and delivers cargo to regulate neuronal morphogenesis is poorly understood, although the cargo-binding kinesin light chain (KLC) subunits contribute to specificity. Our work implicates one such subunit, KLC4, as an essential regulator of axon branching and arborization pattern of sensory neurons during development. Using live imaging approaches in *klc4* mutant zebrafish, we show that KLC4 is required for stabilization of nascent axon branches, proper microtubule (MT) dynamics, and endosomal transport. Furthermore, KLC4 is required for proper tiling of peripheral axon arbors: in *klc4* mutants, peripheral axons showed abnormal fasciculation, a behavior characteristic of central axons. This result suggests that KLC4 patterns axonal compartments and helps establish molecular differences between central and peripheral axons. Finally, we find that *klc4* mutant larva are hypersensitive to touch and adults show anxiety-like behavior in a novel tank test, implicating *klc4* as a new gene involved in stress response circuits.

## Editor's evaluation

Using zebrafish as an in vivo model, this study reveals for the first time that mutations in the kinesin light chain gene klc4, which are known to cause a form of early onset hereditary spastic paraplegia in human, affect specific aspects of neuronal development and nervous system functions. High-resolution movies of developing sensory neurons in vivo and behavioral assays support the key findings that the motor subunit Klc4 plays essential functions in the control of neuronal morphogenesis and compartmentalization as well as associated behaviors.

## Introduction

Proper development of complex neuronal morphology and assembly of neural circuits requires tight regulation of axon growth and branching. Precise control over axon branch formation is essential not only for development but also for plasticity and regeneration after injury. Branching allows one neuron to innervate numerous targets, and in many cases, neurons extend multiple axons or branches

that navigate to distinct targets. Many molecular signaling pathways can influence branching (*Gallo, 2011*; *Kalil and Dent, 2014*), although the mechanisms that control branch formation are still not fully understood. Highly arborized vertebrate sensory neurons, including Rohon-Beard (RB) neurons, serve as an ideal model to study mechanisms of branching and neuronal morphogenesis. Sensory neurons must extend separate axon branches to targets in the periphery and to the CNS to relay sensory information. Peripheral and central axons have different capacity for branching (*Andersen et al., 2011*), and behave differently when contacting other axons—central axons fasciculate with one another while peripheral axons show contact repulsion (*Liu and Halloran, 2005*; *Sagasti et al., 2005*). Contact repulsion of peripheral axons is necessary for establishment of proper axon arbor patterns that 'tile' with arbors from other neurons to effectively innervate the sensory field. Despite the importance of establishing axons with distinct character and branching patterns, the mechanisms regulating molecular compartmentalization of separate axons from one neuron are poorly understood.

Molecular compartmentalization of neurons depends on precisely controlled transport of vesicle and organelle cargos to specific cell locations. Cargos are transported by motor proteins that move in a directional manner along MT tracks. Kinesin-1, the founding member of the kinesin superfamily of motor proteins, is a heterotetramer composed of homodimers of kinesin heavy chains (KHCs), which contain the MT binding and ATPase motor domains, and of kinesin light chains (KLCs), which interact with cargos. Kinesin-1 can mediate long-distance, anterograde transport of multiple cargos including mitochondria, synaptic vesicle precursors, endosomes, lysosomes, and RNA granules (*Hirokawa et al., 2010*; *Morfini et al., 2016*). Kinesin-1 also has roles in organizing MT tracks. The forces it exerts can physically influence the MT lattice (*Shima et al., 2018*; *Triclin et al., 2021*). It can also cross-link MTs by combined interactions of the motor domain and a second MT binding site in the KHC tail, and slide MTs along one another, thereby influencing cell shape (*Lu and Gelfand, 2017*). While it is clear that kinesin-1 is essential for proper development of the nervous system, the mechanisms by which it achieves cell-context specificity of its multiple functions and how its cargo-binding and MT-organizing functions are regulated have yet to be fully defined.

The specific activities of kinesin-1 in a given context are likely governed in part by KLCs. KLCs are known to mediate binding between the kinesin motor and cellular cargos, and also to activate the motor by releasing KHC autoinhibition (*Morfini et al., 2016*; *Wong et al., 2010*; *Woźniak and Allan, 2006*). Vertebrates have four *klc* genes (*klc1-4*), with *klc1, 2* and *4* expressed in the nervous system, but our knowledge of the different roles played by individual KLCs in neurons remains very limited. In *Drosophila*, mutants for the single *klc* gene are lethal in larval stages and have defects in neural development (*Gindhart et al., 1998*; *Mochizuki et al., 2011*), indicating essential neural functions for KLC. Growing evidence supports the idea that molecular functions are divided among vertebrate KLCs and that each has unique roles. Structural and biochemical studies show that KLC1 and KLC2 have differing affinities for cargo adaptor proteins such as JIP1 and JIP3 (*Cockburn et al., 2018*; *Pernigo et al., 2018*; *Zhu et al., 2012*). Research to date has begun to define distinct neurophysiological functions of KLC1 and KLC2. Studies of mouse *klc1* mutants revealed functions for KLC1 in regulating motor behavior (*Rahman et al., 1999*), transport of amyloid precursor protein into axons (*Kamal et al., 2000*), phagosome transport in retinal pigment epithelium (*Jiang et al., 2015*), and axon guidance and cannabinoid receptor transport during neural development (*Saez et al., 2020*). Mouse *klc2* mutants have hearing loss, cochlear hair cell loss and mislocalized mitochondria in hair cells (*Fu et al., 2021*). A study using a peptide inhibitor of KLC2 phosphorylation showed that KLC2 regulates synaptic plasticity, AMPAR trafficking and mood associated behavior (*Du et al., 2010*). Mutations in human *klc2* can cause SPOAN syndrome, which is characterized by progressive spastic paraplegia, optic atrophy and neuropathy with onset in infancy (*Bazvand et al., 2021*; *Melo et al., 2015*). On the whole, previous work supports the idea that individual KLCs have unique, indispensable functions, but has only begun to reveal the specific cellular functions for individual KLCs.

In contrast to KLC1 and KLC2, KLC4 has been the subject of few studies. Mutation of *KLC4* in humans causes a type of hereditary spastic paraplegia (HSP) that manifests in early childhood (*Bayrakli et al., 2015*), indicating essential developmental functions. To date, the only cellular function shown for KLC4 is that it supports mitochondrial function and confers resistance to radiation in cancer cells (*Baek et al., 2018*). A clear cellular function for KLC4 in neurons and therefore insight into the *KLC4* mutation-related pathology is lacking.

In this study, we reveal critical functions for KLC4 in sensory axon branching and axon arbor patterning during development, and in regulation of animal behavior. Using live imaging in *klc4* mutant zebrafish, we found that KLC4 is required for stabilization of nascent peripheral sensory axon branches. The branching defects do not reflect a general growth defect, as axons grew faster in *klc4* mutants and did not show signs of early degeneration. Live imaging also showed that *klc4* mutant peripheral axons have defects in mutual repulsion and instead often fasciculate with one another, a behavior characteristic of central axons, suggesting KLC4 may contribute to the molecular compartmentalization that defines differential behavior of these axons. Using high-speed, in vivo imaging of endosomal vesicle transport and MT dynamics, we showed that *klc4* mutant axons have altered vesicle transport and MT dynamics. Nascent branches in *klc4* mutants have less acetylated tubulin, suggesting less stable MTs. Finally, we investigated the ramifications of KLC4 loss on larval and adult behavior. Larval *klc4* mutant zebrafish show hyperactivity in response to touch, implying heightened sensitivity. Adult *klc4* mutants exhibit increased anxiety-like behavior, indicating essential roles for KLC4 in neural circuit formation and/or function in adults.

## Results

### *klc4* is expressed during early development

*klc* genes encode proteins with conserved functional domains including the heptad repeat region that mediates binding to KHC, six cargo-binding tetratricopeptide repeats (TPRs), a C-terminal lipid-binding amphipathic helix, and an LFP motif that mediates KLC auto-inhibition (*Figure 1A*; *Antón et al., 2021*; *Cockburn et al., 2018*; *Morfini et al., 2016*; *Pernigo et al., 2018*; *Zhu et al., 2012*). As a first step toward understanding developmental functions of KLC4, we investigated the expression of zebrafish *klc4* at different stages of development. Zebrafish *klc4* has two alternative splice isoforms, one long and one short, both of which are also present in other vertebrates including mice and humans. In zebrafish, the long *klc4* transcript encodes a full length protein of 620 amino acids while the short isoform is predicted to encode a 185 amino acid protein, ending just before the LFP motif (*Figure 1—figure supplement 1A*). To determine whether both transcripts are expressed during development, we performed non-quantitative reverse transcriptase-PCR with transcript-specific primers on RNA extracted from embryos during early development (*Figure 1B*, *Figure 1—figure supplement 1A*). We found that both isoforms are expressed throughout early development, with expression detectable at 2 hours post fertilization (hpf) and continuing through the initial period of neuronal axon outgrowth, pathfinding and branching, between 16 and 28 hpf (*Figure 1B*). We next used in situ hybridization to determine the spatial expression pattern of *klc4* during development, using a riboprobe that binds both transcripts. At 24 hpf, *klc4* is expressed in clusters of neurons in the telencephalon, epiphysis, diencephalon, ventral midbrain and hindbrain, and in the trigeminal and the posterior lateral line (pLL) ganglia (*Figure 1C and D*). In the spinal cord, *klc4* is strongly expressed in the Rohon-Beard sensory neurons, and at lower levels in other neuron populations (*Figure 1E*). At 48 hpf, *klc4* continues to be expressed broadly in the nervous system with high expression levels in the trigeminal ganglia, pLL ganglia, and RB neurons (*Figure 1F, G and H*). A riboprobe that only binds the long isoform shows a similar expression pattern to the probe that recognizes both isoforms (*Figure 1I, J and K*), indicating the full-length *klc4* mRNA is present in developing neurons. The early expression of *klc4* in diverse neural populations in both the central and peripheral nervous system suggests it has functions in neural development.

### Generation of *klc4* mutants

To assess the role of KLC4 during neural development, we used CRISPR/Cas9 to generate *klc4* mutants. We used a guide RNA targeting exon 3 of *klc4*, which encodes a part of the N-terminal heptad repeat region. One resulting allele, *klc4^uw314^*, has a 25 bp deletion in exon 3 from 404bp to 428bp in the heptad repeat region that leads to an early stop after 9 missense amino acids (*Figure 1A*, *Figure 1—figure supplement 1B*). Heterozygous *klc4^uw314^* mutants were outcrossed to wild type (AB strain) fish for three generations to remove any off-target mutations from the background. In situ hybridization analysis of homozygous *klc4^uw314^* mutant embryos showed virtually no *klc4* mRNA expression, suggesting that the mutant transcript undergoes nonsense-mediated decay (*Figure 1L, M and N*).

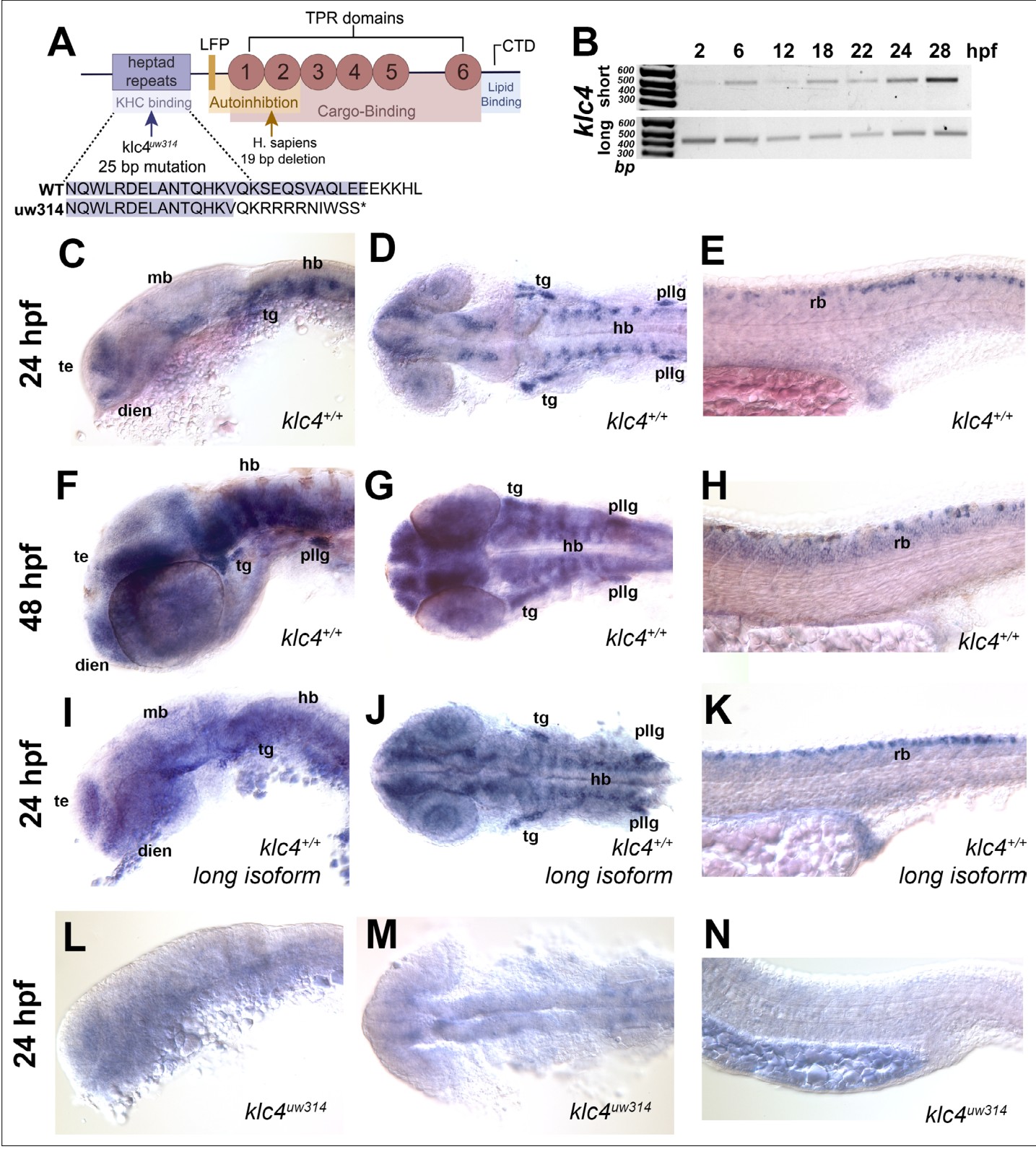

**Figure 1.** *klc4* mRNA is expressed in developing zebrafish embryos. (**A**) Diagram of the highly conserved domains of KLC4. Arrows indicate where mutation occurs in humans and in our zebrafish mutant. (**B**) Qualitative RT-PCR showing both long and short *klc4* isoforms are expressed throughout development. (**C–D**) In situ hybridization for *klc4* mRNA shown in lateral view (**C**) or dorsal view (**D**) of the head in wild type 24 hpf embryos. (**E**) Lateral view of trunk showing *klc4* expression in Rohon-Beard sensory neurons in a 24 hpf wild type embryo. (**F–G**) *klc4* expression in the lateral view (**F**) or dorsal view (**G**) of the head of a 48 hpf wild type embryo. (**H**) Lateral view of trunk showing *klc4* expression in the spinal cord of a 48 hpf wild type

*Figure 1 continued on next page*

*Figure 1 continued*

embryo. (**I–K**) Lateral (**I,K**) and dorsal (**J**) views of *klc4* long isoform specific in situ hybridization in wild type 24 hpf embryos. Lateral (**L,N**) and dorsal (**M**) views of *klc4* in situ hybridization in 24 hpf *klc4^uw314^* mutant embryo head (**L,M**) and trunk (**N**) showing little expression of *klc4*. Te = telencephalon, dien = diencephalon, mb = midbrain, hb = hindbrain, tg = trigeminal ganglion, pllg = posterior lateral line ganglion, rb = rohon beard neuron.

The online version of this article includes the following source data and figure supplement(s) for figure 1:

**Source data 1.** Original.tiff and.scn file for the gel of RT-PCR results showing presence of *klc4* isoforms.

**Figure supplement 1.** PCR strategy for the detection of long and short *klc4* isoforms, and sequence alignment of wild type and *klc4uw314* mutant.

## *klc4^uw314^* mutants show reduced branching of RB peripheral sensory axons

Because *klc4* is highly expressed in spinal RB sensory neurons, we first analyzed the effects of *klc4* mutation on their axon growth and morphology. We crossed the *klc4^uw314^* mutants with a transgenic line in which RB neurons are labeled with membrane-targeted GFP [*Tg(–3.1ngn1:gfp-caax)*] (*Andersen et al., 2011*) and used light sheet microscopy to image RB axon development across the whole embryo (*Figure 2A and B*). Inspection of mutant embryos revealed that RB peripheral axon branches appeared less dense and displayed less overall innervation of the sensory field compared to wild type (*Figure 2A and B*, *Figure 2—videos 1 and 2*). To quantify these differences we labeled sensory neurons with anti-HNK1 antibody at 24 hpf (*Figure 2C, D and E*). We counted the number of axons that crossed eight regions of interest drawn at standardized intervals from the central axon fascicle, proceeding from dorsal to ventral (*Figure 2—figure supplement 1*). Consistent with our visual impression of reduced branching, mutant embryos had consistently fewer axons at each ROI (*Figure 2F*). The reduction in branches at the most dorsal levels in the mutant embryos may indicate an effect on initiation of the peripheral axon, which most often forms as a branch from the central axon near the cell body (*Andersen et al., 2011*).

As an alternative approach to understand the role of KLCs in axon branching, and to develop a more time-resolved means to disrupt KLC function, we took a pharmacological approach. Specifically, embryos were treated with Kinesore, a small molecule that has been shown to interfere with the interaction between the KLC TPR repeat domain and cargo adaptor proteins, while also relieving KLC auto-inhibition and activating kinesin-1's ability to cross-link MTs and regulate MT dynamics (*Randall et al., 2017*). We found that treatment of wild type embryos with 100 µM Kinesore during early stages of peripheral axon arborization, from 17 hpf to 24 hpf, caused a reduction in axon branch number in ventral regions at 24 hpf of a similar magnitude to that seen in *klc4^uw314^* mutants (*Figure 2G-I*), although more dorsal branching appears unaffected. These results give further evidence that properly regulated KLC-cargo interactions or kinesin-1-MT interactions during axon outgrowth are important for successful axon branching.

## KLC4 is required for stabilization of axon branches

To investigate which specific dynamic processes during branching are affected by *klc4* loss of function, we used live confocal imaging. Zebrafish RB axons branch by two mechanisms: growth cone bifurcation (GCB) and interstitial back branching (IB) (*Gibson and Ma, 2011*). To determine if GCB, IB or both were defective in mutants, we used the *Tg(–3.1ngn1:gfp-caax)* transgenic line (*Andersen et al., 2011*). Z-stacks of RB peripheral axon arbors were collected every 1–1.5 min for 2–4 hr beginning from approximately 18 hpf. The number of branch initiation and retraction events during a 2-hr time window were counted and categorized as either a GCB (*Figure 3A*, white arrows) or IB event (*Figure 3A*, magenta arrows). Surprisingly, we found no difference in the rate of branch initiation between *klc4^uw314^* mutant and wild-type embryos (*Figure 3C*). In both conditions, new branches were initiated via GCB more frequently than IB, but there was no difference in the proportion of GCB versus IB branching events between wild-type and *klc4^uw314^* mutants.

Nevertheless, the live imaging immediately suggested an explanation for the branch number deficit: failure of branches to persist. That is, following formation, while most of the branches in wild type animals were relatively stable and continued growing (*Figure 3—video 1*), those in the mutants often retracted after formation (*Figure 3B*, *Figure 3—video 2*). Quantification showed that in *klc4^uw314^* mutants, 57.7% of GCB branches and 61.9% of IB branches retracted after initiating, compared to 14.2% of GCB and 11.8% of IB branches in wild type (*Figure 3D*). Mutants showed no

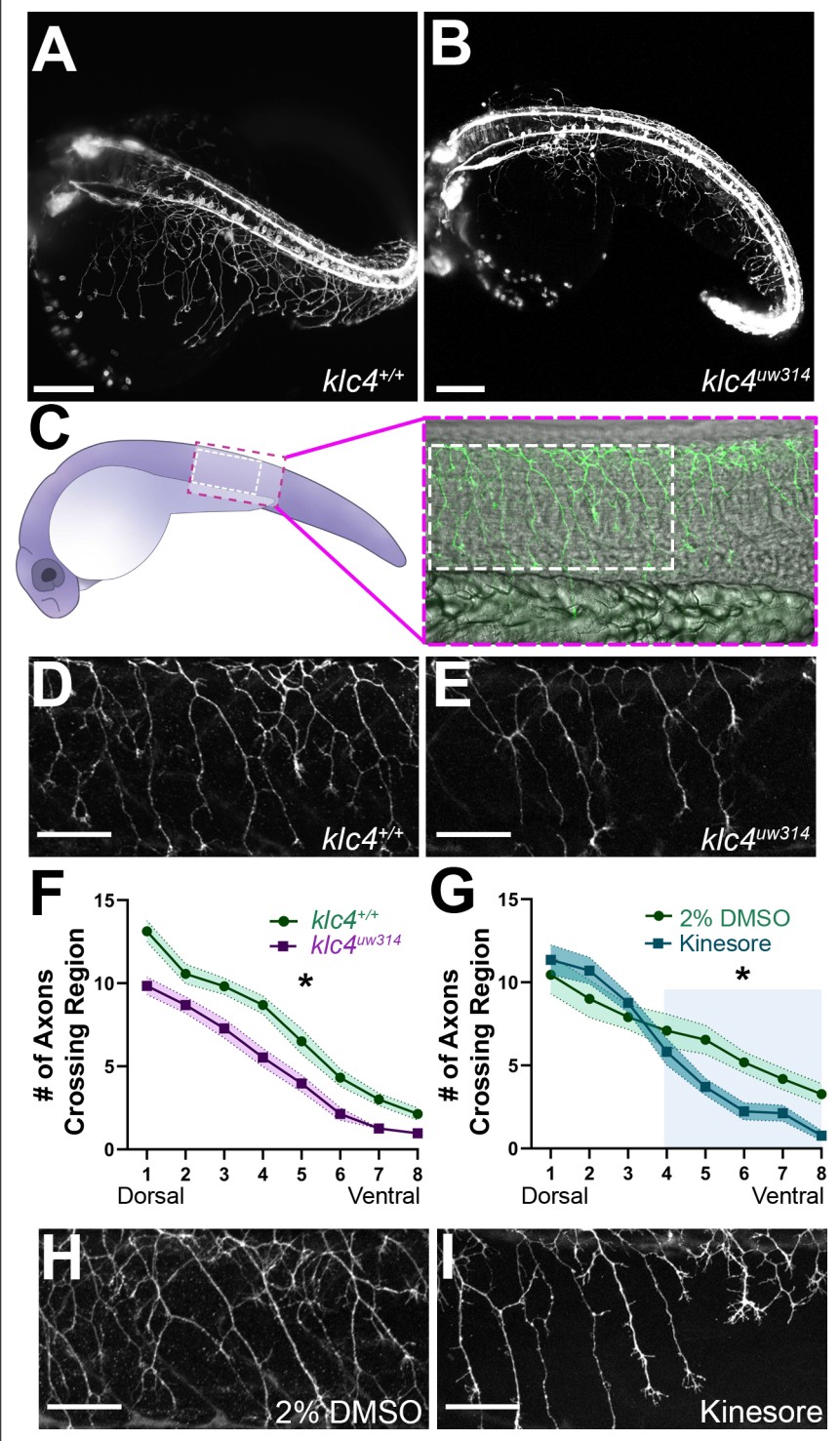

**Figure 2.** *klc4^{uw314}* mutants have reduced Rohon-Beard peripheral axon branching. (**A–B**) Stills taken at approximately 24 hpf from long term light sheet movies of wild type (**A**) or *klc4^{uw314}* mutant (**B**) embryo development Scale bar = 150μm. (**C**) Diagram of a 24 hpf embryo indicating the area over the yolk tube extension imaged for axon analysis, with the standard region used for analysis outlined by the white dashed rectangle. (**D–E**) Representative examples of HNK-1 stained embryos showing peripheral axon branching in wild type (**D**) and *klc4^{uw314}* I embryos. (**F–G**) Quantification of axon branching across three technical replicates in wild type (n=32) and *klc4^{uw314}* mutant (n=32) embryos (**F**) and DMSO (n=12) and kinesore (n=17) treated embryos (**G**). Error bars = SEM. Statistical significance was measured by comparing the area under the curve for (**F–G**) using t-test. For

*Figure 2 continued on next page*

*Figure 2 continued*

(**F**), *p=0.0378 for entire graph. For (**G**), *p=0.0484 for shaded region. (**H–I**) Representative examples of HNK-1 stained embryos showing peripheral axon branching in DMSO (**H**) and kinesore treated (**I**) embryos. Scale bars for D,E,H,I = 50 μm.

The online version of this article includes the following video, source data, and figure supplement(s) for figure 2:

**Source data 1.** The number of branches crossing a selected region of interest (1-8) for wild type and *klc4^uw314^* mutant embryos.

**Source data 2.** The number of branches crossing a selected region of interest (1-8) for wild type embryos treated with either 2% DMSO or 100 μM Kinesore.

**Figure supplement 1.** Schematic describing our branch analysis method.

**Figure 2—video 1.** A wild type Tg(–*3.1ngn1:gfp-caax*) embryo imaged by light sheet microscopy over 11.6 hr to show sensory axon development, from approximately 18 to 29.6 hpf.
https://elifesciences.org/articles/74270/figures#fig2video1

**Figure 2—video 2.** A *klc4^uw314^* mutant Tg(–*3.1ngn1:gfp-caax*) embryo imaged by light sheet microscopy over 8.6 hr to show sensory axon development, from approximately 19 to 27.6 hpf.
https://elifesciences.org/articles/74270/figures#fig2video2

---

significant directional bias to the retracted branches (on average, 52.5% of branches lost per neuron were anteriorly-directed, and 47.2% were posteriorly-directed. p=0.6588). This phenotype cannot be explained as a general axon growth defect, as *klc4^uw314^* mutant axons grew at a faster rate than wild type axons (*Figure 3E*). Thus, these data suggest that KLC4 is required specifically for stabilization of axon branches.

## Sensory axon arborization pattern is altered in *klc4^uw314^* mutants

Peripheral sensory axons develop a distinctive arborization pattern to successfully innervate the sensory field. During this process, axons mutually repel each other upon contact (*Sagasti et al., 2005*; *Liu and Halloran, 2005*). Precise regulation of contact repulsion as well as branching are critical for effective tiling of the entire skin. To determine whether mutants have altered arborization and to quantify branching pattern in an unbiased manner, we developed an image analysis pipeline to measure axon directional orientation and overall axon abundance (*Figure 4*). We used OrientationJ (*Rezakhaniha et al., 2011*) to measure the fluorescence signal of aligned pixels at each orientation (*Figure 4A and B*) and created an axon directionality "profile" that represents the average signal at each 1.0 degree angle interval across multiple embryos (*Figure 4C*). In wild-type embryos, most axons are oriented in ventral-posterior and ventral-anterior directions, with a higher proportion of axon signal in posteriorly directed angles, peaking approximately between –50° and –85° (*Figure 4A and C*, *Figure 4—figure supplement 1A*). Little signal was detected in the horizontal (0 degrees) orientation. In contrast, *klc4^uw314^* mutants had reduced overall axon signal compared to their wild-type cousins (*Figure 4B and C*, *Figure 4—figure supplement 1B*), consistent with the reduced branch density. Mutants showed an overall directional profile similar to wild type (*Figure 4B and C*). However, because differences in axon orientation between wild type and mutants may be masked by the overall difference in signal, we normalized the data and displayed it as a percentage of total signal at each angle. We found a small but significant increase in axons oriented in the posterior direction in *klc4^uw314^* mutants, specifically those oriented between –20.5° and –50.5° (*Figure 4D*). There was no difference in the proportion of anterior-directed axons between wild type and *klc4^uw314^* mutants (*Figure 4—figure supplement 1C*).

   To investigate what changes in axon behavior could generate the posterior orientation bias in mutants, we turned to live imaging. We analyzed the growth direction of actively extending axons by measuring the orientation angle of the distal-most axon shaft at each time point over 2–4 hr movies (*Figure 5A*) and displaying the distribution of growth angles as rose plots. We found *klc4^uw314^* mutant axons had a preference for posterior growth paths, while wild-type axons had no anterior-posterior bias (*Figure 5B and C*). We then measured the angles between newly bifurcating sister branches. Prior to bifurcation, axons commonly grow in a posterior direction in both wild type and mutant embryos (*Figure 5D*). As growth cones bifurcate in wild-type embryos, the sister branches grow away from each other (*Figure 5D' and E*, *Figure 5—video 1*), resulting in an average branch angle of 96.5 degrees 15 min after branch formation, and 91.7 degrees 75 min after branch formation (*Figure 5E and G*).

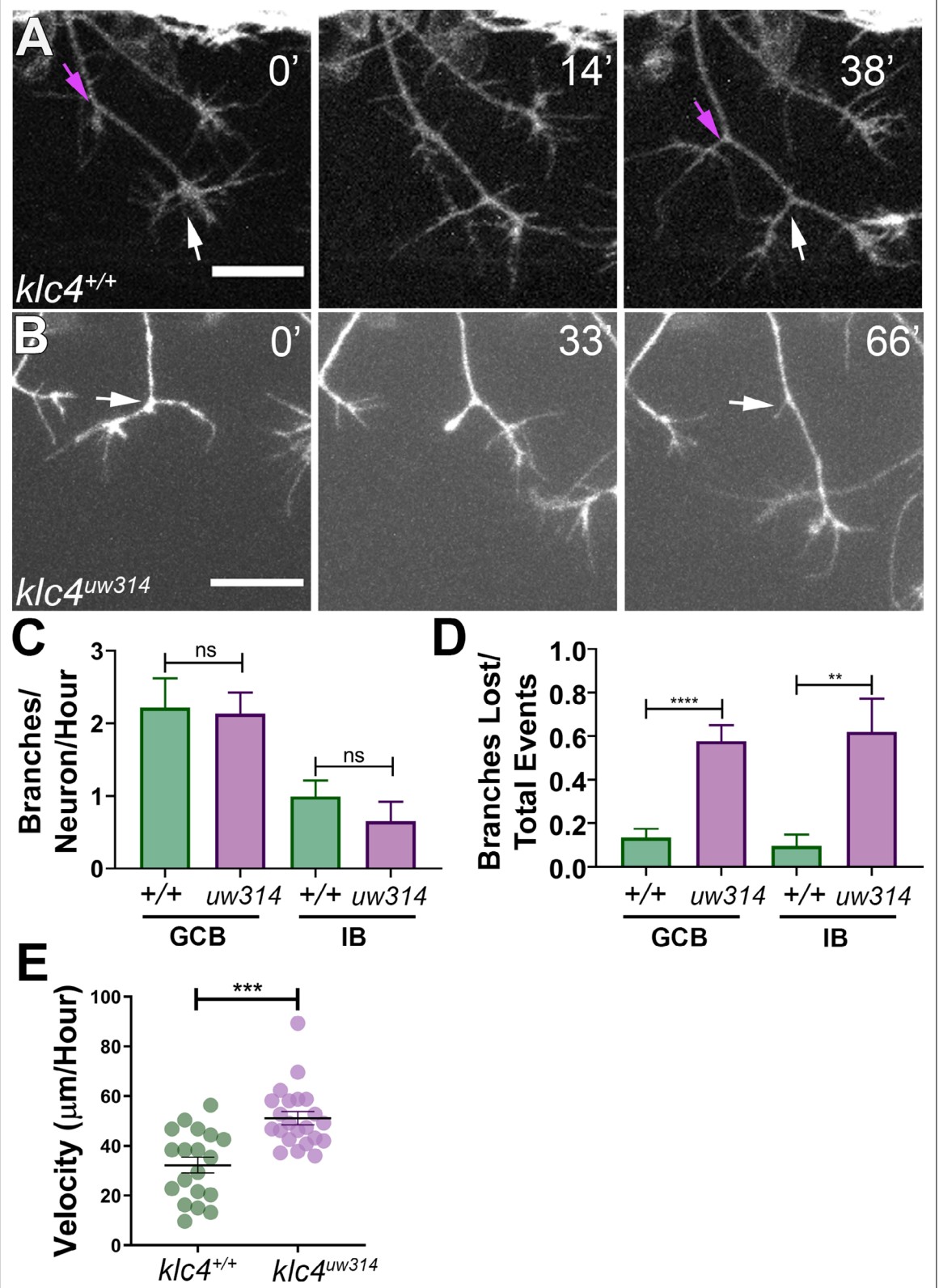

**Figure 3.** Branches initiate but do not stabilize in *klc4^uw314^* mutants. (**A**) A wild type axon forms stable branches by growth cone bifurcation (GCB, white arrows) and interstitial branching (IB, magenta arrows) over the course of 38' min. (**B**) A *klc4^uw314^* mutant axon initiates a branch via GCB (0minutes, white arrow), but one branch begins to retract and is nearly fully gone after 66' min (panel 3, white arrow). 20 μm scale bar. (**C**) Quantification of axon branch initiations per neuron per hour in wild type and mutant embryos wild type GCB mean = 2.22 branches/neuron/hr, N=11 neurons from 4 embryos.

*Figure 3 continued on next page*

*Figure 3 continued*

*Klc4$^{uw314}$* GCB = 2.13 branches/neuron/hr, N=14 neurons from 4 embryos. Wild type IB=0.99 branches/neuron/hr, N=10 neurons from 4 embryos. *Klc4$^{uw314}$* IB = 0.65 branches/neuron/hr, N=8 neurons from 4 embryos. Error bars = SEM. There was no significant difference in branch initiation between wild type and mutant axons. GCB *P*=0.73, IB *P*=0.15, Mann–Whitney test. (**D**) Quantification of branches that retracted after initiation in wild type and *klc4$^{uw314}$* embryos. Data are displayed as a branch loss ratio (branches retracted/total branching events). Branches created by either GCB or IB were both more likely to retract in *klc4$^{uw314}$* embryos (wild type GCB mean = 0.13, N=11 neurons from 4 embryos; *klc4$^{uw314}$* GCB = 0.58, N=14 neurons from 4 embryos; wild type IB = 0.10, N=10 neurons from 4 embryos; *klc4$^{uw314}$* IB = 0.62, N=8 neurons from 4 embryos). **p=0.0055, ****p<0.0001, Mann–Whitney test. Error bars = SEM. (**E**) Quantification of axon growth velocity. Axons in *klc4$^{uw314}$* mutants grow faster than in wild type. Wild type N=19 axons from 3 embryos. *Klc4$^{uw314}$* N=22 axons from 4 embryos. ***P=0.0001, Mann-Whitney test. Error bars = SEM.

The online version of this article includes the following video and source data for figure 3:

**Source data 1.** The number of branches initiated per neuron per hour separated by type of branch (growth cone bifurcation = GCB, Interstitial Branch = IB).

**Source data 2.** The number of branches of a category (growth cone bifurcation = GCB, Interstitial Branch = IB) that retracted divided by the total number of branching events of that category.

**Source data 3.** The growth speed of wild type and *klc4$^{uw314}$* mutant embryos.

**Figure 3—video 1.** Branches in a wild type Tg(*-3.1ngn1:gfp-caax*) embryo growing and spreading over 15 min.

https://elifesciences.org/articles/74270/figures#fig3video1

**Figure 3—video 2.** An example of a growth cone bifurcation in a *klc4$^{uw314}$* mutant Tg(*-3.1ngn1:gfp-caax*) embryo where the left branch begins to retract at approximately 23 min.

https://elifesciences.org/articles/74270/figures#fig3video2

This behavior results in axon arbors with an even distribution of axon orientation angles in wild-type embryos (*Figure 4C*). In *klc4$^{uw314}$* mutants, the newly formed branches did not separate as much, resulting in a shallower average angle of 79.7 degrees between sister branches 15 min after branch formation, and narrowing to 60.2 degrees after 75 min (*Figure 5F and G*, *Figure 5—video 2*). In fact, branches in *klc4$^{uw314}$* mutants at times failed to repel at all, instead appearing to collapse onto one another (see *Figure 6B*). These results suggest that KLC4 may be involved in mediating mutual repulsion between sister axon branches. Moreover, the posterior growth bias detected by automated orientation analysis in *klc4$^{uw314}$* mutant embryos (*Figure 4C*) is likely the result of branch separation failure. Because axons initially tend toward posterior growth, reduced branch separation or collapse of anteriorly directed branches onto the more posterior axon shaft leads to axon arbors with an overall increase in posterior orientation angles. Thus, our automated, population-scale method of axon orientation profiling led to the discovery of a branch separation defect, demonstrating the utility of the profiling method for revealing alterations in axon arbor pattern.

## Peripheral axons in *klc4$^{uw314}$* mutants exhibit abnormal fasciculation behavior

Live imaging also revealed a broader defect in axon repulsion: normally, in wild type embryos when a peripheral axon growth cone contacts another axon (either a branch of the same neuron or of another neuron) it retracts or changes direction to grow away from the other axon, thus minimizing contact between axons (*Sagasti et al., 2005*; *Liu and Halloran, 2005*; *Figure 6A*). However, in *klc4$^{uw314}$* mutants, axons often fasciculate after such contacts and grow together for substantial distances (*Figure 6B and C*, *Figure 6—video 1*). In some cases, axons were fasciculated so tightly that they only became distinguishable as two axons when pulled apart by the force of a developing branch (*Figure 6C*, arrows, *Figure 6—video 1*, minutes 41–47, *Figure 6—video 2*, minutes 24–44). In wild-type embryos, all observed fasciculations (n=3) resolved by retraction or defasciculation, but only 46.5% of fasciculations (n=19) in *klc4$^{uw314}$* mutants were resolved in this manner. Fasciculation behavior was quantified by measuring the frequency and length of fasciculation events occurring during our live imaging period. We found that fasciculation events were rare in wild type embryos but occurred relatively frequently in mutants (*Figure 6D*). Fasciculations ranged from 7.5 to 60.5 μm in length in *klc4$^{uw314}$* mutants (*Figure 6E*). Together, these results show that KLC4 is important for the mutual repulsion needed to shape sensory arbors. The mutual repulsion normally exhibited by peripheral axons is in contrast to the fasciculation behavior of the central axons from the same neuron. The failure of

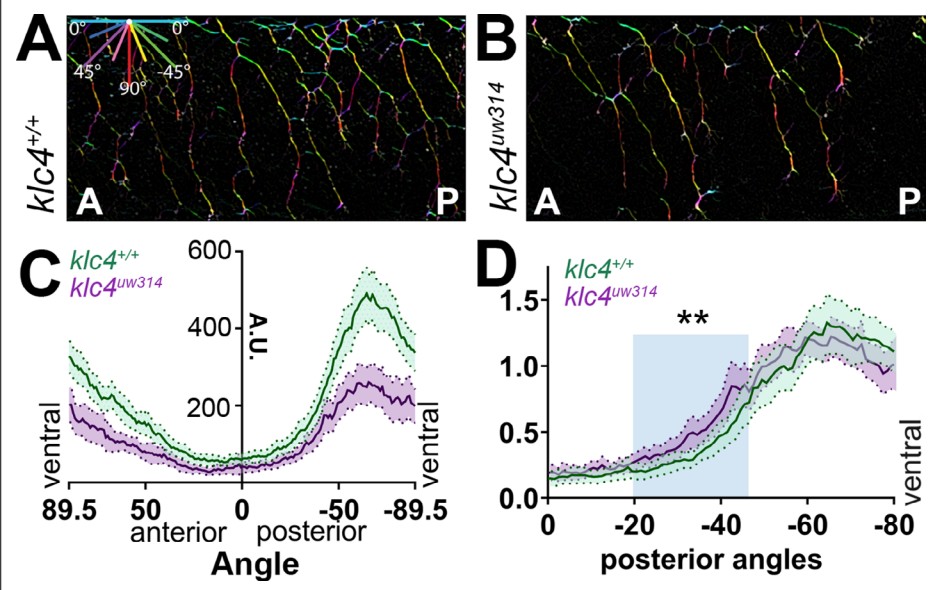

**Figure 4.** Axon orientation profiling reveals increased posterior directionality of peripheral axons in *klc4^uw314^* mutants. (**A–B**) A representative region of axon analysis for a 24 hpf wild type (**A**) and *klc4^uw314^* (**B**) embryo. The axons are color coded for directionality according to the legend in the upper left corner of (**A**). (**C**) Plot of axon directionality in 24 hpf wild type (green) and *klc4^uw314^* mutants (magenta). (**D**) Axon directionality data was normalized to total signal and posterior angles are displayed to compare directional bias. Area under the curve was measured for shaded area and tested for statistical significance using a t-test. **\*\****P*=0.0047. For (**C–D**), error bars = 95% CI. Wild type N=40 embryos, *klc4^uw314^* N=23 embryos.

The online version of this article includes the following source data and figure supplement(s) for figure 4:

**Source data 1.** Histogram of the axon signal at each angle for wild type and *klc4^uw314^* mutant axons, used to generate directional distribution plots.

**Source data 2.** Histogram data in Source Data 1 that has been normalized to the total axon signal present in the sample.

**Figure supplement 1.** Additional data for branch directionality profiling.

*klc4^uw314^* mutant peripheral axons to show repulsion suggests that KLC4 may be involved in the localization of factors controlling central versus peripheral axon behavior.

## RB neurons have aberrant apical axons in *klc4^uw314^* mutants

RB neurons in *klc4^uw314^* mutants also showed abnormal morphology suggestive of altered cell polarity. We found that some RB neurons in *Tg(–3.1ngn1:gfp-caax);klc4^uw314^* mutant embryos and Kinesore-treated embryos projected a process from the apical cell body that crossed the midline within the dorsal spinal cord (*Figure 7A and B*, *Figure 7—videos 1 and 2*). Small apical protrusions sometimes form on RB neurons, especially in the anterior portion of the spinal cord, though they can be found nearly anywhere along the spinal cord. These processes are typically short, dynamic, and are most common early in development (18–24 hpf; *Figure 7—video 3*). In wild-type embryos, they rarely cross the midline of the spinal cord. We counted the number of midline crossing apical processes from images of anti-HNK-1-labeled embryos captured in the spinal cord overlying the yolk extension and found that 4% of neurons (n=1,509 neurons in 51 embryos) in *klc4^uw314^* mutants had midline crossing processes compared to 1% in wild type embryos (n=608 neurons in 21 embryos) (*Figure 7C*). Similar results were obtained following treatment with Kinesore from 18 to 24 hpf, with 7% of Kinesore-treated neurons (n=433 neurons in 17 embryos) extending an apical protrusion that crossed the midline, compared to 2% of neurons in vehicle-treated control (n=436 neurons in 15 embryos) (*Figure 7C*). To further assess neuronal morphology and the pathways of the aberrant projections, we sparsely labeled individual RB neurons by injecting DNA encoding TagRFP-caax driven by the –3.1ngn1 promoter at the one-cell stage. This mosaic cell labeling showed that the apical processes grew to and fasciculated with the contralateral central axon fascicle in both *klc4^uw314^* mutant and kinesore-treated embryos

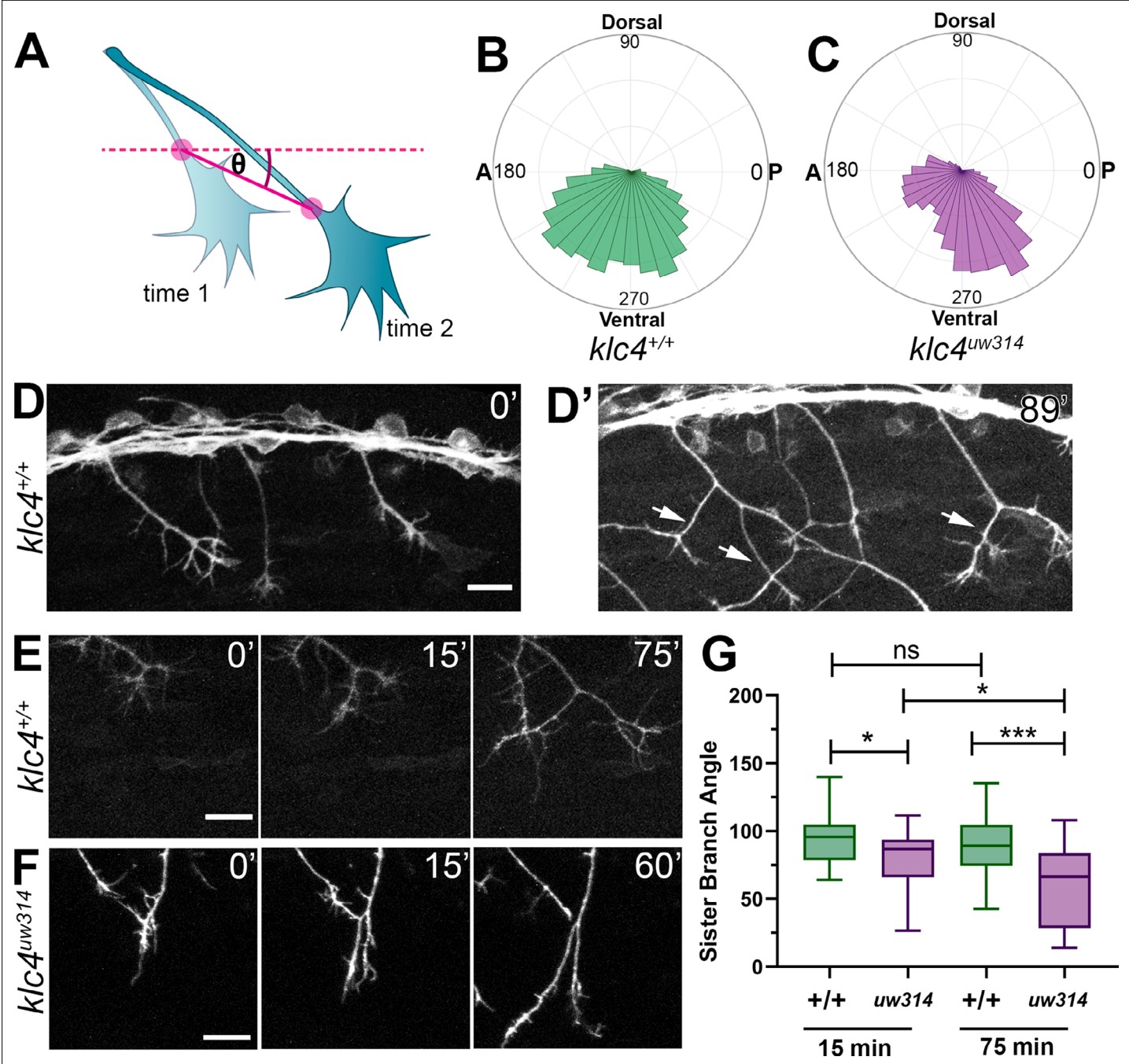

**Figure 5.** Live imaging shows that *klc4uw314* mutants have a bias toward posterior growth direction and defective sister branch separation. (**A**) Illustration of how axon growth direction was measured. The angle between an axon's position in each frame and the frame preceding it was measured. (**B–C**) These angles were plotted as rose plots to show growth direction distributions. Wild type axons grow evenly between anterior and posterior directions, while *klc4uw314* mutant axons show a preference for posterior directed growth. Wild type N=19 axons from 3 embryos. *Klc4uw314* N=22 axons from 4 embryos. (**D**) Stills from a live capture of wild type axon outgrowth, showing that axons often grow posteriorly, but anterior directed branches form over time (D', white arrows). (**E**) As a branch forms in a wild type peripheral axon, the angle between sister branches remains consistent as axons grow apart. (**F**) In *klc4uw314* mutants, sister branches do not grow as far apart and the sister branch angle narrows over time compared to wild type. All scale bars = 20 μm. (**G**) Quantification of the angle between newly formed sister branches at 15 and 75 min after branch creation. Wild type 15' N=37 branches, *klc4uw314* 15' N=36 branches, Wild type 75' N=25 branches, *klc4uw314* 75' N=18 branches. 4 embryos per genotype. *p=0.021, ***p=0.0003, One-way ANOVA. Error bars = 5-95th percentile.

The online version of this article includes the following video and source data for figure 5:

**Source data 1.** The rose plot data generated with the Chemotaxis Tool for wild type axon tracking data.

*Figure 5 continued on next page*

*Figure 5 continued*

**Source data 2.** The rose plot data generated with the Chemotaxis Tool for *klc4^uw314^* mutant axon tracking data.

**Source data 3.** The angle between sister branches, measured at 15 min after branch formation and 75 min after branch formation.

**Figure 5—video 1.** Example of a wild type peripheral axon growing out while maintaining a wide branch angle between sister branches.

https://elifesciences.org/articles/74270/figures#fig5video1

**Figure 5—video 2.** Peripheral axon growth in a *klc4^uw314^* mutant embryo demonstrating both fasciculation between axons from the same neuron (self-fasciculation) and the narrowing of the angle between sister axons over time.

https://elifesciences.org/articles/74270/figures#fig5video2

(*Figure 7D and F*), suggesting they may respond to cues directing central axon behavior despite emerging from an ectopic apical location. These results suggest that KLCs have a role in determining the location of axon emergence from the cell body.

## Endosomal vesicle transport is altered in *klc4^uw314^* mutants

To begin exploring the mechanisms by which KLC4 affects axon arbor development, we imaged endosomal vesicle transport. We previously found that a properly functioning endosomal trafficking system is necessary for RB axon branching (*Ponomareva et al., 2014*; *Ponomareva et al., 2016*). Moreover, we showed that the kinesin adaptor Clstn1 is required for branching and that it functions in part by trafficking Rab5-labeled endosomal vesicles to branch points (*Ponomareva et al., 2014*). We used a previously described zebrafish Rab5c construct (*Clark et al., 2011*) to label Rab5-containing endosomes in vivo. We expressed GFP-Rab5c in individual RB neurons by injecting *ngn1:GFP-Rab5c* DNA into embryos at the one-cell stage (*Figure 8A*). We imaged vesicle movement using a high-speed swept field confocal microscope that allows imaging of rapid events in 3D. We captured z-stacks of RB axon arbors at 2–5 s intervals for 7–10 min periods and used kymography to quantify endosome dynamics (*Figure 8A′*). We found some alterations in transport. In *klc4^uw314^* mutants, a smaller proportion of vesicles moved in the anterograde direction compared to wild type (*Figure 8B*), and the maximum anterograde vesicle run length was reduced (*Figure 8C*), suggesting that some aspects of Rab5 vesicle transport or a subset of Rab5 vesicles rely on KLC4 function. However, there was no significant difference in velocity of the Rab5 vesicles that were moving, suggesting these vesicles can attach to the motor without KLC4 (*Figure 8D*).

To determine whether KLC4 regulates Rab5 vesicular transport from the cell body specifically to axon branch points, we labeled Rab5c with photoactivatable GFP (PA-GFP). We co-injected *ngn1:PA-GFP-Rab5c* and *ngn1:TagRFP-caax* to label individual RB neurons. We photoactivated the GFP in the cell body during stages when peripheral axons are actively growing and branching, and captured images at ~1–2 min intervals for 30–40 min after photoactivation (*Figure 8—figure supplement 1A*). We previously reported that in wild type embryos, PA-GFP-Rab5c vesicles originating in the cell body appeared to accumulate at the peripheral axon initiation point and in secondary branch points within minutes of photoactivation (*Ponomareva et al., 2014*). To quantify vesicle accumulations at branch points over time, we divided the movies into 4 time periods (t1-t4), made maximum intensity projections of the z-stacks for t1-t4, and measured the fluorescent signal at branch points as a percentage of the signal at the first time period (t1). In both wild type and *klc4* mutant neurons, the intensity increased similarly during this period, suggesting KLC4 does not play a significant role in vesicle delivery from the cell body to branch points (*Figure 8—figure supplement 1B*).

## Microtubule dynamics are altered in *klc4^uw314^* mutants

Because local changes in MT dynamics are associated with axon branch formation, and MT invasion of nascent axon branches is required for branches to be stabilized and maintained (*Yu et al., 1994*; *Dent and Kalil, 2001*; *Yu et al., 2008*; *Gallo, 2011*; *Armijo-Weingart and Gallo, 2017*), we assessed whether the MT cytoskeleton is affected in *klc4^uw314^* mutants. To visualize MT dynamics in vivo, we used the MT plus-tip binding protein EB3 fused to GFP (*Stepanova et al., 2003*). EB3-GFP binds to the plus ends of actively polymerizing MTs, which appear as moving GFP puncta or 'comets'. We expressed EB3-GFP in RB neurons by injecting *ngn1:EB3-GFP* DNA, and imaged with a swept field confocal microscope. We captured z-stacks at 2–5 s intervals for periods ranging from 7 to 10 min and analyzed microtubule dynamics using kymography (*Figure 9A*, *Figure 9—video 1*). In axons,

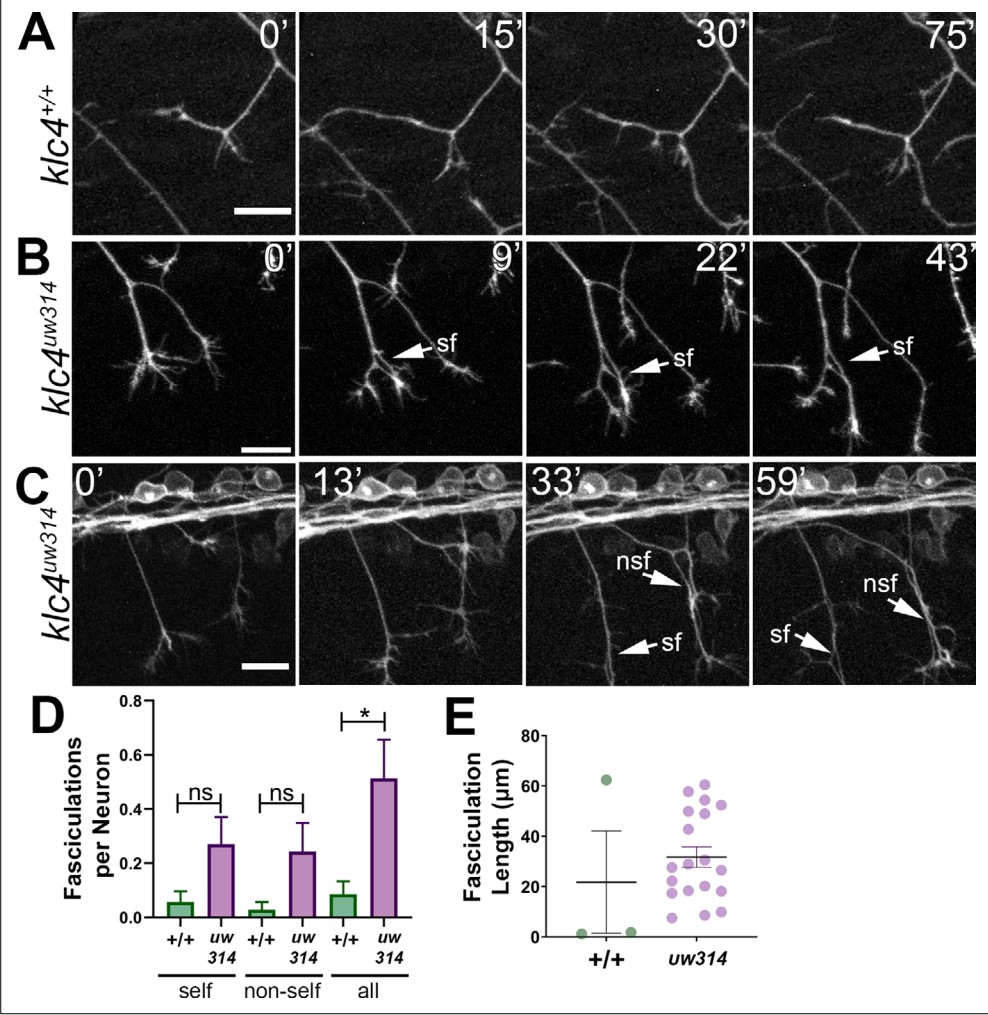

**Figure 6.** Loss of KLC4 results in aberrant axon fasciculation. (**A**) In wild type embryos, the most common response to self or non-self contact between axons is retraction. Two neighboring axons in a wild type embryo make contact (time 0') and within 30 min, one of the axons has retracted. (**B–C**) When axons make contact in *klc4*[uw314] embryos, they often fasciculate instead of retracting. This happens with both self-contacts (**B–C, sf**) and non-self-contacts (**C**), nsf. All scale bars = 20 µm. (**D**) Fasciculations per neuron, separated by self or non-self, as well as all fasciculations combined. Wild type N=35 neurons in 4 embryos. *Klc4*[uw314] N=37 neurons in 5 embryos. Error bars = SEM. (**E**) Fasciculation length was measured from the last frame of each 2 hr imaging period. Wild type N=3 fasciculations, *klc4*[uw314] N=19 fasciculations. Mean with SEM displayed over individual data points. Because fasciculation events are rare in wild-type embryos and a length measurement is conditional on the event occurring, we could not apply a statistical test to assess the significance of the difference in fasciculation length.

The online version of this article includes the following video and source data for figure 6:

**Source data 1.** Fasciculations per neuron, separated as self or non-self, and combined (all).

**Source data 2.** Fasciculation length measured in microns.

**Figure 6—video 1.** A peripheral axon in a *klc4*[uw314] mutant embryo contacts a neighboring peripheral axon and fasciculates with it.

https://elifesciences.org/articles/74270/figures#fig6video1

**Figure 6—video 2.** Tight fasciculation is revealed in a *klc4*[uw314] mutant peripheral axon as a gap opens between the two axons as an interstitial branch forms.

https://elifesciences.org/articles/74270/figures#fig6video2

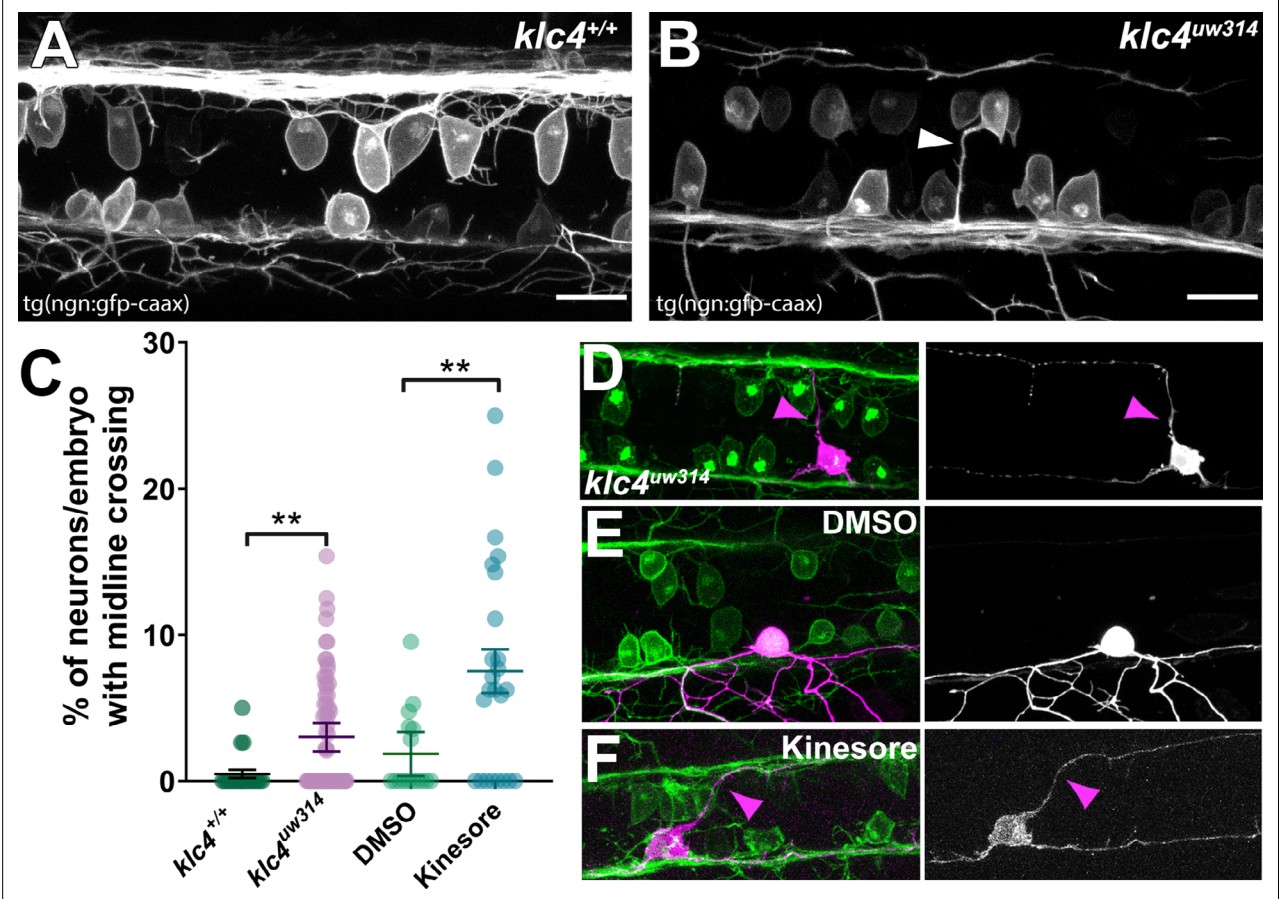

**Figure 7.** Midline crossing of apical protrusions occurs more frequently when KLC function is disrupted. (**A–B**) Confocal projections (dorsal views) of partial z-stacks containing only the optical sections in the plane of the RB cell bodies within the spinal cord in a wild-type Tg(–3.1ngn1:gfp-caax) embryo (**A**), or a *klc4uw314* Tg(–3.1ngn1:gfp-caax) embryo (**B**), with an apical protrusion from a neuron crossing the midline and reaching the contralateral side (white arrowhead). Scale bars = 20 µm. (**C**) The percentage of neurons per embryo that had an apical protrusion crossing the midline of the spinal cord was quantified. **p<0.0052, Mann–Whitney test. Mean with SEM displayed over individual data points. Wild type N=608 neurons from 21 embryos, *klc4uw314* N=1496 neurons from 47 embryos, 2% DMSO N=429 neurons from 15 embryos, Kinesore N=595 neurons from 21 embryos. (**D–F**) Mosaic labeling of individual neurons by injection with –3.1ngn1:gfp-caax DNA into *klc4uw314* mutants (**D**), DMSO-treated embryos I or kinesore-treated embryos (**F**) showing that apical protrusions (magenta arrowheads) can fasciculate onto the contralateral central axon.

The online version of this article includes the following video and source data for figure 7:

**Source data 1.** The percentage of neurons per embryo that had an apical protrusion that fully crossed the midline of the spinal cord.

**Figure 7—video 1.** Light sheet microscopy of an embryo treated with 100µM Kinesore, showing what appear to be apical protrusions crossing the midline of the spinal cord.
https://elifesciences.org/articles/74270/figures#fig7video1

**Figure 7—video 2.** High-speed, high-magnification light sheet microscopy of a *klc4uw314* mutant embryo showing an apical protrusion growing across the midline with a dynamic tip that shows growth-cone like behavior.
https://elifesciences.org/articles/74270/figures#fig7video2

**Figure 7—video 3.** A cropped portion of *Figure 2—video 1* showing normal behavior of apical protrusions in the spinal cord of a wild-type embryo.
https://elifesciences.org/articles/74270/figures#fig7video3

the majority of comets move in the anterograde direction, reflecting the plus-end-distal organization of MTs. We found no change in the percentage of retrograde comets between wild type and *klc4uw314* mutants, suggesting KLC4 is not necessary for organization of MT polarity (*Figure 9B and C*). However, we found a significant increase in the frequency of anterograde comets in peripheral axons, but not central axons, in *klc4uw314* mutant RB neurons (*Figure 9B and D*, *Figure 9—figure supplement 1A*), suggesting increased MT dynamics in peripheral axons. In addition, we found that the velocity of EB3-GFP comets, both anterograde and retrograde, was increased in peripheral and

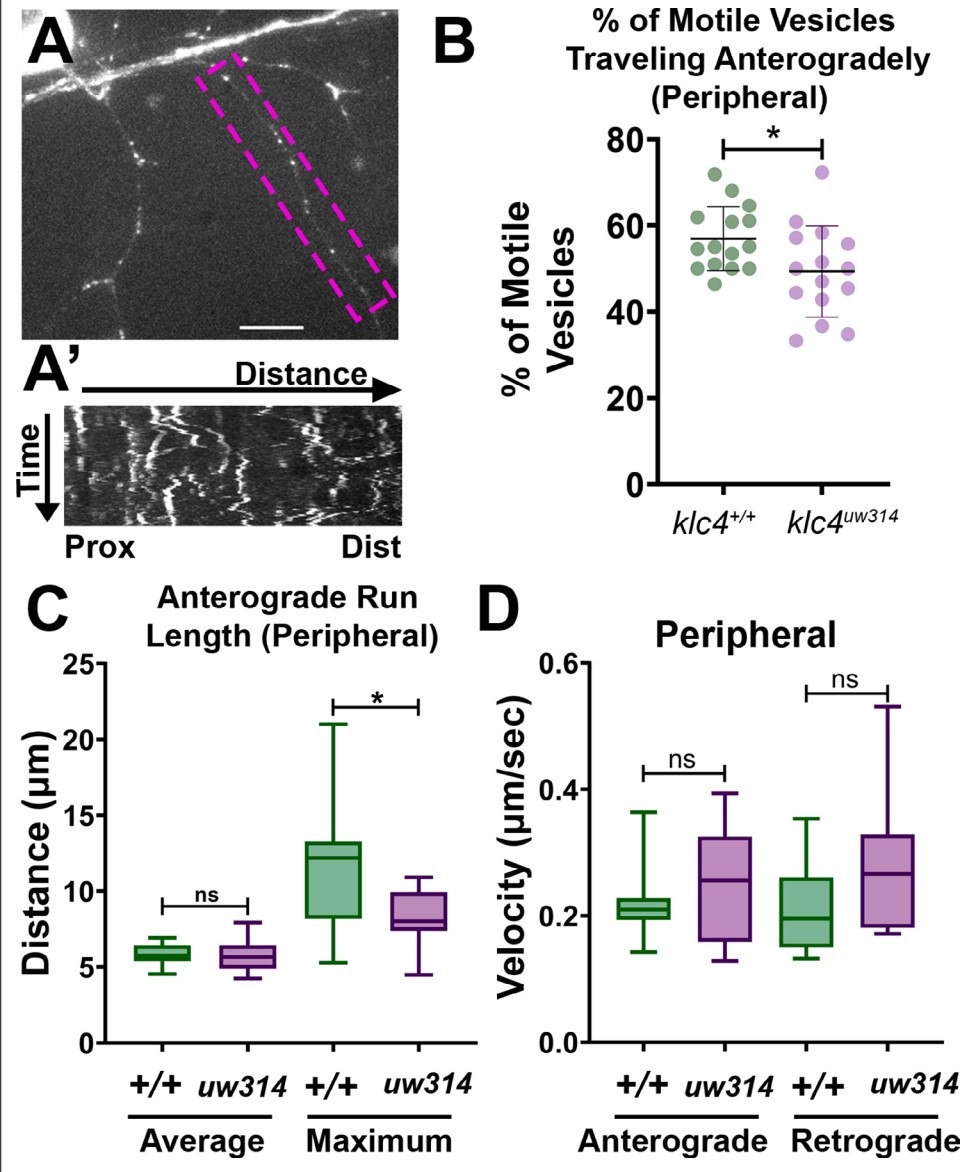

**Figure 8.** Endosomal trafficking dynamics are disrupted in *klc4³¹⁴* mutants. (**A**) Live RB neuron expressing eGFP-Rab5c. Scale bar = 10 μm. (**A'**) Representative kymograph of axon segment boxed in magenta in (**A**).(**B**) Percentage of anterogradely moving vesicles out of all motile Rab5 vesicles in peripheral axons. The proportion of anterogradely moving vesicles is decreased in *klc4³¹⁴* mutants, *p=0.0372 by Mann-Whitney test; error bars = mean with SD. Wild type N=15 embryos, *klc4³¹⁴* N=15 embryos. (**C**) Average Rab5 vesicle run length and maximum anterograde vesicle run length in peripheral axons. *p=0.0358, ns p=0.5359, Mann-Whitney test. (**D**) Rab5 vesicle velocity in peripheral axons. NS by Mann-Whitney test (anterograde p=0.6505, retrograde p=0.0792), wild type N=12 embryos, *klc4³¹⁴* N=11 embryos. For all data, one neuron per embryo was analyzed. Error bars in box and whisker plots represent 5-95th percentile.

The online version of this article includes the following video, source data, and figure supplement(s) for figure 8:

**Source data 1.** The percentage of motile vesicles per cell that were traveling anterogradely in peripheral RB axon branches.

**Source data 2.** The distance traveled by Rab5 comets, including both average and maximum run length.

**Source data 3.** The measured velocity of Rab5 comets in peripheral axons.

**Figure supplement 1.** Vesicles labeled with photoactivatable Rab5 accumulate normally at branch points.

**Figure 8—video 1.** An RB neuron expressing photoactivatable Rab5 (green) and membrane-bound TagRFP (magenta).

https://elifesciences.org/articles/74270/figures#fig8video1

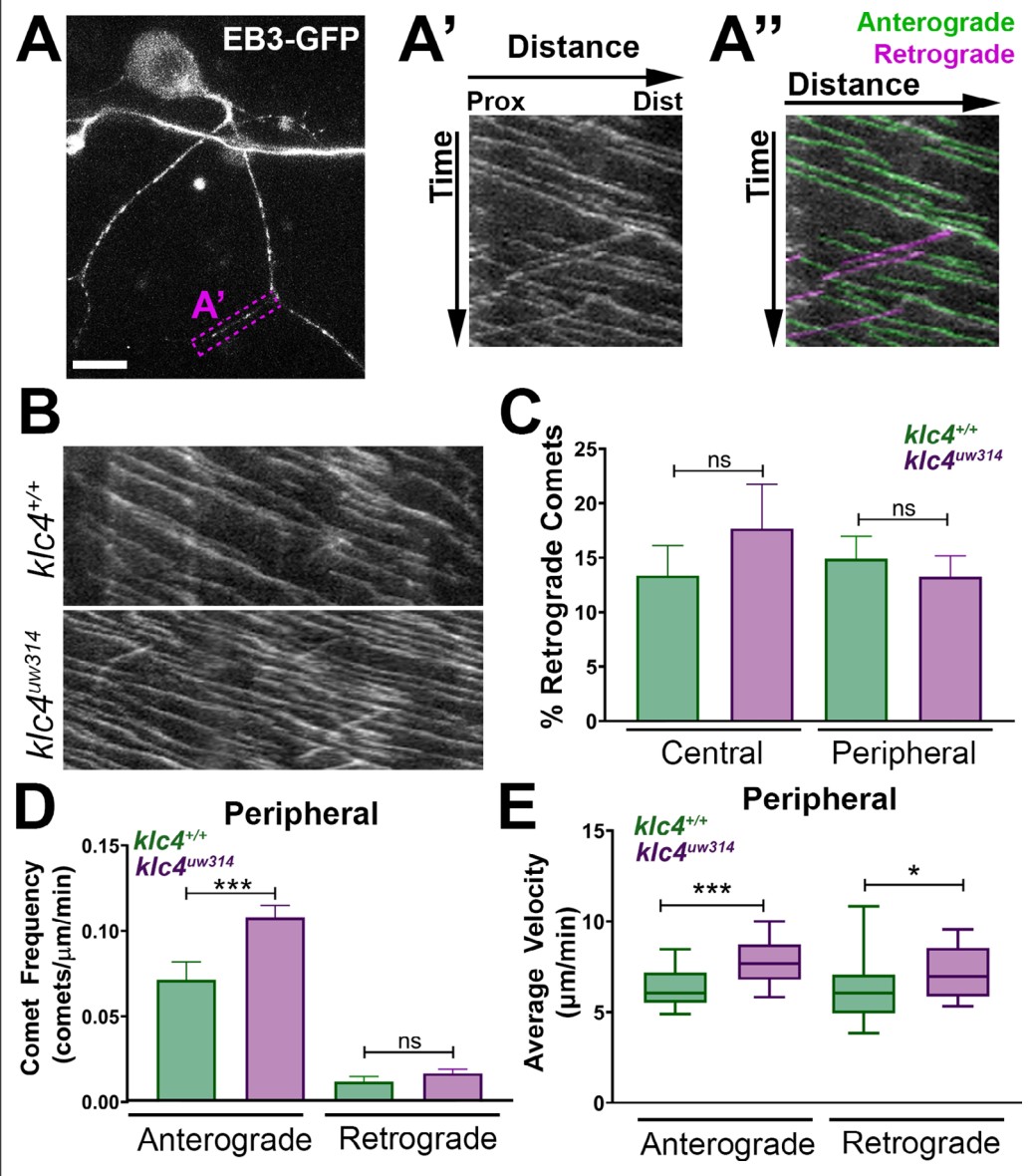

**Figure 9.** Microtubule dynamics are increased in *klc4^uw314* mutants. (**A-A"**) Live RB neuron expressing EB3-GFP, scale bar = 10 μm. Magenta box in (**A**) indicates axon segment selected for kymograph in (**A'**). (**A"**) EB3 comets identified as anterogradely (green) or retrogradely (magenta) directed based on their angle. (**B**) Representative kymographs from peripheral axon segments of wildtype (top) or *klc4^uw314* (bottom). (**C**) Quantification of the percentage of retrograde comets out of all comets imaged for both peripheral and central axons. NS by Mann-Whitney test. (**D**) Peripheral axon comet frequency. ***p=0.0003, Mann-Whitney test. (**E**) Peripheral axon comet velocity. ***p=0.0004, *p=0.042, Mann-Whitney test. For all data, error bars = SEM. Wild type central N=8 segments, *klc4^uw314* central N=15 segments, Wild type peripheral N=26 segments, *klc4^uw314* peripheral N=48 segments. Wild type data from 10 neurons in 7 embryos. *Klc4^uw314* data from 13 neurons in 8 embryos. All comets in a kymograph were measured.

The online version of this article includes the following video, source data, and figure supplement(s) for figure 9:

**Source data 1.** Source Data 1: All measurements of EB3-GFP comets in wild type and *klc4^uw314* mutants, averaged per measured segment.

**Figure supplement 1.** Additional quantification of microtubule dynamics in wild type and *klc4* mutant axons.

(**A**) Central axon comet frequency. NS by Mann-Whitney test. (anterograde p=0.4664, retrograde p=0.7268) (**B**) Central axon comet velocity. *p<0.05, Mann-Whitney test. (**C**) Average distance traveled by peripheral axon comets. NS by Mann-Whitney test (anterograde p=0.1476, retrograde p=0.9288) (**D**) Average distance traveled by

*Figure 9 continued on next page*

*Figure 9 continued*

central axon comets. NS by Mann-Whitney test (anterograde p=0.3738, retrograde p=0.8836) (**E**) Average comet run time for peripheral axon comets. NS by Mann-Whitney test (***p=0.0001, NS p=0.1389) (**F**) Average comet run time for central axon comets. NS by Mann-Whitney test (anterograde p=0.3363, retrograde p=0.3196) For all data, error bars = SEM. Wild type central N=8 segments, *klc4*$^{uw314}$ central N=15 segments, Wild type peripheral N=26 segments, *klc4*$^{uw314}$ peripheral N=48 segments. Wild type data from 10 neurons in 7 embryos. *Klc4*$^{uw314}$ data from 13 neurons in 8 embryos.

**Figure 9—video 1.** An RB neuron expressing EB3-GFP to tag plus ends of microtubules.
https://elifesciences.org/articles/74270/figures#fig9video1

central axons in *klc4*$^{uw314}$ mutants (***Figure 9E***, ***Figure 9—figure supplement 1B***). There was no difference in the distance traveled by EB3-GFP comets in any condition (***Figure 9—figure supplement 1C***,D), although duration of anterograde comet runs was reduced in peripheral axons, consistent with their increased velocity (***Figure 9—figure supplement 1E***,F). Thus, the length of new MT polymer growth at plus ends is not affected in mutants, but mutant MTs polymerize faster, suggesting a role for KLC4 in regulating the growth rate of MT plus ends.

## MTs in nascent branches are less stable in *klc4*$^{uw314}$ mutants

To ask whether MT dynamics are altered in newly forming branches, we co-injected *ngn:EB3-GFP* DNA with *ngn:TagRFP-caax* DNA to label cell membranes, and imaged EB3-GFP comets in nascent branches (***Figure 10A***). We found no difference in the frequency of comets in nascent branches between wild type and mutants (***Figure 10B***). EB3 comet frequency also did not differ between mutant and wild type when separated by branch type (IB vs GCB, data not shown, p>0.7 by t-test). However, because EB3-GFP comet frequency in peripheral axons overall is increased in *klc4*$^{uw314}$ mutants, the equivalent comet frequencies between mutant and wild type in nascent branches may represent a net reduction in MT polymerization into nascent branches in mutants.

As another approach to assess MT stability in new branches, we fixed embryos at a stage of active branching (24 hpf) and double-labeled neurons with anti-HNK1 and an antibody to acetylated tubulin, a modification found on stable MTs (***Webster and Borisy, 1989***; ***Figure 10C and D***). We analyzed terminal branches up to 10 µm in length formed by interstitial branching or bifurcation, then scored those with acetylated tubulin signal. We found a significant reduction in the percentage of nascent branches with acetylated tubulin in *klc4*$^{uw314}$ mutants (***Figure 10E***). This finding suggests that KLC4 is required for MTs to stabilize in newly formed branches.

## Larval and adult *klc4*$^{uw314}$ fish show behavioral defects

The formation of functional neural circuits ultimately depends on the precise orchestration of cell biological processes during neural development. To ask whether KLC4 loss affects circuit function, we analyzed larval escape behavior in response to touch, a behavior mediated by sensory neurons. Larva at 3 days post-fertilization (dpf) were given a light tail touch, evoking a swim response (***Figure 11A***). The time of the swim response and number of swim bouts completed was measured for each larva. *Klc4*$^{uw314}$ mutant larva swam for a longer total time (wild type median = 0.65 s, *klc4*$^{uw314}$ 1.49 s) and performed more swim bouts (wild type median = 1 bout, *klc4*$^{uw314}$ mutants = 2 bouts) after touch stimulus than wild type larva (***Figure 11B and C***), suggesting KLC4 loss leads to hypersensitivity to touch.

The cell biological defects found during development in *klc4*$^{uw314}$ mutants seemed likely to lead to deficits in adult fish as well. Indeed, we found that *klc4*$^{uw314}$ mutants are adult viable and fertile but are smaller in size than their wild-type siblings (***Figure 11—figure supplement 1A***). To measure body size in a controlled manner, we in-crossed F4 homozygous *klc4*$^{uw314}$ mutants from both AB and mixed AB/TL backgrounds and separately in-crossed their wild-type (AB) siblings. The F5 embryos were born on the same day, raised side-by-side in an identical environment, and fed the same diet. We measured the body length of these fish at 7 weeks and at 9 weeks (***Figure 11—figure supplement 1B***). *klc4*$^{uw314}$ mutants were on average 83.3% the size of their wild-type cousins at 7 weeks. This difference remained at 9 weeks, despite both the wild type and *klc4*$^{uw314}$ mutant fish increasing in size, with *klc4*$^{uw314}$ mutant fish measured at 84.6% the length of their wild-type cousins. Smaller body size has also been reported in *klc1* mutant mice (***Rahman et al., 1999***), supporting the idea that KLCs are important for post-embryonic growth of the animal.

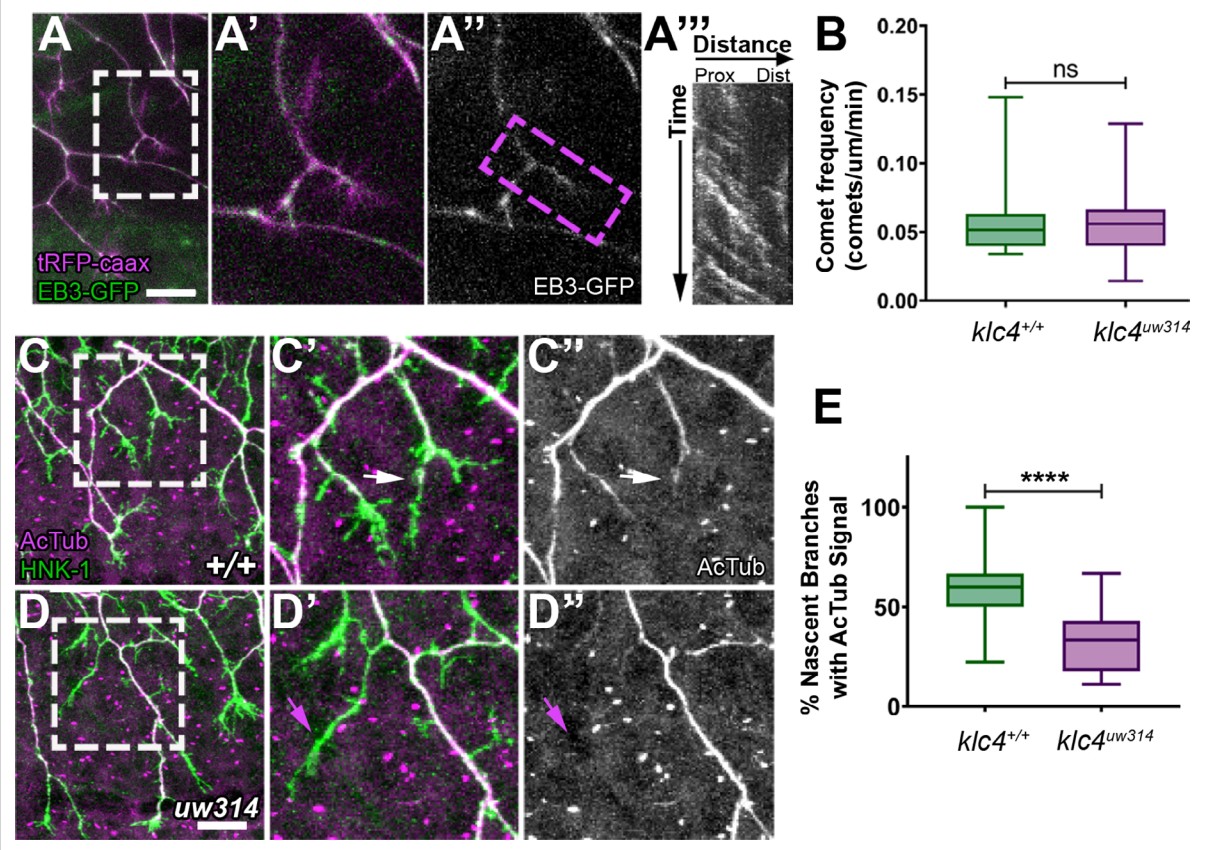

**Figure 10.** Nascent RB axon branches show reduced microtubule stabilization in *klc4uw314* mutants. (**A**) Still image of nascent RB branches expressing EB3-GFP in a live wild type embryo at 23hpf. (**A'**) Zoom in of the branches outlined in white in (**A**).(**A"**) EB3-GFP signal in nascent branches in (**A'**). (**A'''**) Representative kymograph generated from nascent branch outlined in magenta in (**A'**). (**B**) Quantification of peripheral axon comet frequency. NS by Mann-Whitney test, p=0.8410. Wild type N=22 nascent branches from 9 embryos, *klc4uw314* N=25 nascent branches from 8 embryos. (**C-D"**) Representative examples of HNK-1 and acetylated tubulin stained embryos at 24hpf. White arrows in (**C'-C"**) indicate nascent branches with tubulin acetylation. Magenta arrows in (**D'-D"**) indicate nascent branches lacking tubulin acetylation. (**E**) Percentage of nascent branches containing acetylated tubulin out of all nascent branches. Fewer nascent branches contained acetylated tubulin in *klc4uw314* mutants. ****p<0.0001 by Mann-Whitney test. Wild type N=132 nascent branches from 15 embryos, and *klc4uw314* N=143 nascent branches from 16 embryos. Scale bars = 20 µm. Error bars in box and whisker plots represent 5-95th percentile.

The online version of this article includes the following source data for figure 10:

**Source data 1.** Source Data 1: The EB3 comet frequency in nascent axons.

**Source data 2.** The percentage of nascent branches with discernable acetylated tubulin signal out of all nascent branches observed.

Multiple neural systems are affected in the human disease caused by *KLC4* mutation; patients show defects in vision, hearing, movement and cognition (*Bayrakli et al., 2015*). Thus, we analyzed adult behavior to more broadly investigate KLC4's roles in neural circuit function. To ask if the mutant's small size reflected a defect in feeding behavior, we assayed feeding ability (*Howe et al., 2018*). Adult zebrafish were isolated in individual 1 L tanks, then 50 larval zebrafish were added to each tank. The fish were allowed to feed on the larvae for one hour, then removed from the tank to allow counting of the remaining larvae. Wild type fish were effective feeders, eating on average 90% of available zebrafish larvae (*Figure 11—figure supplement 1C*). In comparison, *klc4uw314* mutants ate on average only 60% of the larvae available. We noted that the mutant fish that performed the worst in the feeding assay underwent bouts of freezing behavior during the assay. Upon introduction of the larvae to the tank, the fish ceased movement for extended periods.

Freezing behavior in zebrafish in response to a novel environment has been previously described and interpreted as increased anxiety (*Wong et al., 2010*; *Maximino et al., 2010*; *Ziv et al., 2013*; *Khan et al., 2017*). To test whether *klc4uw314* mutant fish show anxiety-like behavior, we used a novel tank diving test, which is similar in principle to the open field test used in rodents (*Wong et al., 2010*;

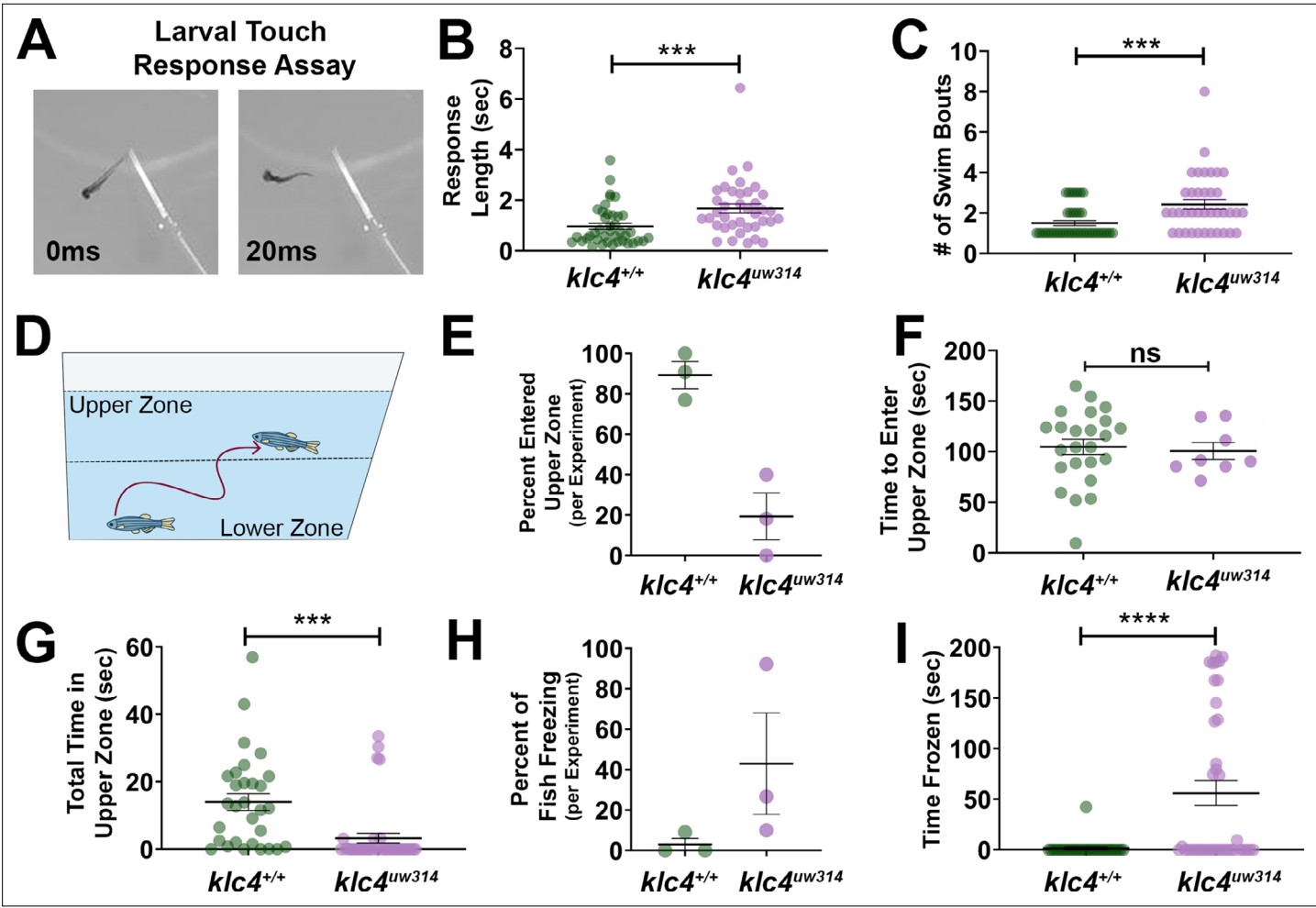

**Figure 11.** Larval and adult *klc4*<sup>uw314</sup> mutants have behavioral defects (**A**) Example of touch assay administration. A 3 dpf larva is touched on the tail using a gel loading tip, provoking a swim response. (**B**) Larval response time was measured starting from beginning of swim to cessation of motion. (**C**) The number of swim bouts performed by each larva during total post-touch swim time was counted. For B-C, each point represents an individual larva. ***p=0.0005, Mann-Whitney test. Error bars = SEM. Wild type N=45, *klc4*<sup>uw314</sup> N=41. 55 total larva were tested for each genotype, but larva that did not respond were not included in these data. (**D**) Schematic of the novel tank test design, with upper and lower zones demarcated by the center dotted line. (**E**) Percentage of fish per experiment that entered the upper zone of the tank during a 3-min exposure to a novel tank. Error bars = SEM. Each point represents one experiment day: wild type N=3 experiments with 30 total fish, *klc4*<sup>uw314</sup> N=3 experiments with 39 total fish. (**F**) The time it took for individual fish to first cross into the upper zone during a 3-min exposure to a novel tank. Points represent individual fish, error bars = SEM. Wild type N=24, *klc4*<sup>uw314</sup> N=8. Fish that did not cross into the upper zone are not represented in this data. NS by t-test, p=0.7766. (**G**) The total time fish spent in the upper zone of the novel tank during a 3-min exposure. Points represent individual fish, ***p=0.0002 by t-test, error bars = SEM. Wild type N=30, *klc4*<sup>uw314</sup> N=39. (**H**) Percentage of fish to freeze per experiment during a 3-min exposure to the novel tank. Error bars = SEM. Each point represents one experiment day: wild type N=3 experiments with 30 total fish, *klc4*<sup>uw314</sup> N=3 experiments with 39 total fish. (**I**) The total time fish spent frozen during a 3-min exposure to a novel tank. Points represent individual fish, ****p<0.0001 by Mann-Whitney test, error bars = SEM. Wild type N=30, *klc4*<sup>uw314</sup> N=39.

The online version of this article includes the following source data and figure supplement(s) for figure 11:

**Source data 1.** The length of time each individual larva spent swimming after a touch stimulus (non-responding fish removed from graph).

**Source data 2.** The number of swim bouts performed by each individual larva after a touch stimulus (non-responding fish removed from graph).

**Source data 3.** The average percentage of fish that entered the upper zone of the tank per experimental trial.

**Source data 4.** The time it took an individual fish to first cross into the upper zone of the tank.

**Source data 5.** The total time spent by each individual fish in the upper zone throughout the experimental time period.

**Source data 6.** The average percentage of fish that performed one or more freezing bouts per experimental trial.

**Source data 7.** The total time spent frozen by each individual fish throughout the experimental time period.

**Figure supplement 1.** Loss of *klc4* results in reduced size and ineffective feeding in adult zebrafish.

*Maximino et al., 2010*). In this test, fish are introduced to an unfamiliar tank and allowed to explore. Zebrafish typically prefer to stay in the lower zone of the tank initially, and once they become acclimated, they begin to explore the upper regions of the tank (*Figure 11D*). We introduced adult animals (7 months old) to a novel tank and filmed their behavior for 3 min. Analysis revealed that 90.9% of wild type fish explored the upper zone of the tank during the 3-min window, compared to only 18.8% of the *klc4<sup>uw314</sup>* mutant fish (*Figure 11E*). Of the animals that did explore the upper zone, both wild type and *klc4<sup>uw314</sup>* mutant animals took an average of 100 s before their first cross into the upper zone (*Figure 11F*). Wild type animals spent an average of 14 s in the upper region of the tank, compared to just 3.26 s for *klc4<sup>uw314</sup>* mutants (*Figure 11G*). Notably, freezing behavior, defined for this experiment as cessation of all movement except for gills for at least 3 s, was observed in an average of 43% of mutant fish, and only 3% of wild type fish (*Figure 11H*). 15.4% of *klc4<sup>uw314</sup>* mutant fish, but no wild type animals, froze for the full 3-min duration of the experiment (*Figure 11I*).

Studies in both zebrafish and rodents have shown that repeated exposure to the unfamiliar environment leads to habituation of the anxiety behavior. In zebrafish, habituation manifests as increased

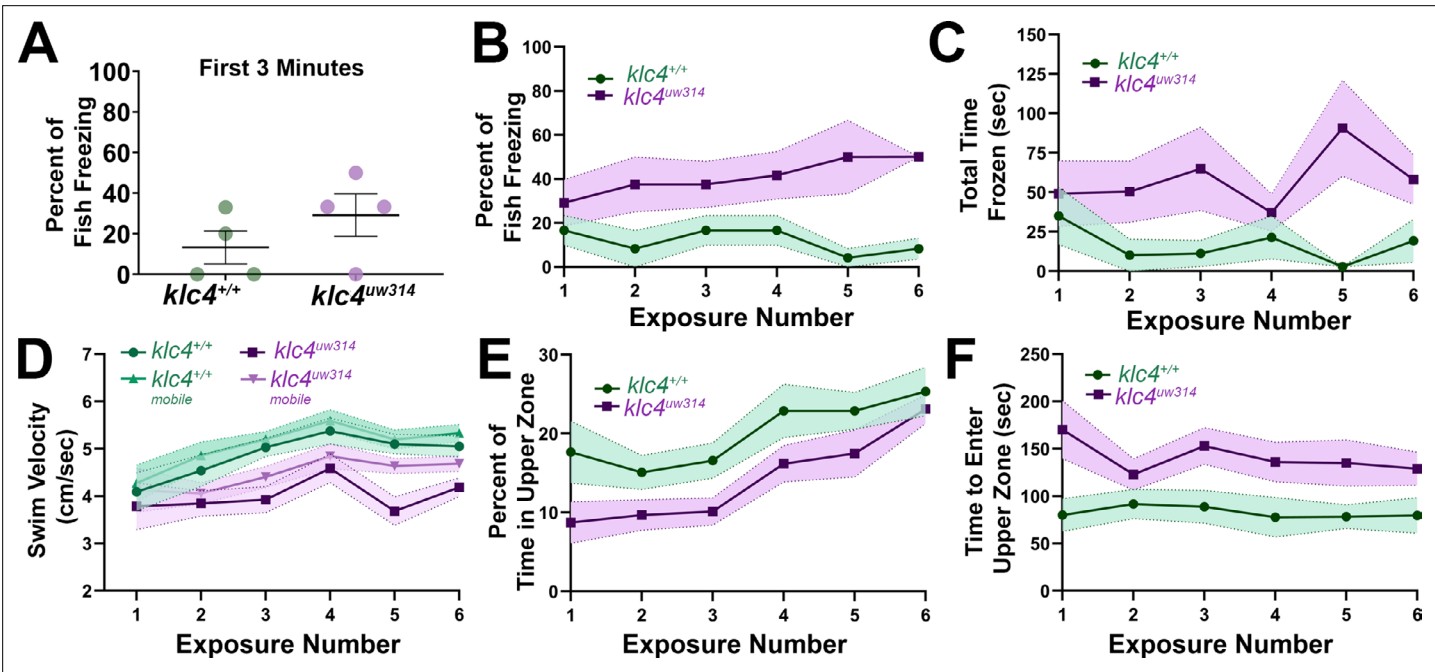

**Figure 12.** Adult *klc4<sup>uw314</sup>* mutant anxiety-like behavior does not fully habituate after repeat exposure to a novel environment. (**A**) Percentage of fish per experiment that froze during the first 3 min of a 10-min initial exposure to a novel tank. (**B**) Plot of the percentage of fish that froze during the course of six 10-min exposures to the same novel tank. (**C**) Plot of the average time spent frozen during the course of six 10-min exposures to the same novel tank. (**D**) Plot of the average total and mobile-only swim velocities of fish during each of six 10-min exposures to the same novel tank. Mobile-only velocity excludes periods of time the fish spent at a velocity of 0 cm/s (frozen). (**E**) Plot of the percentage of time that fish spent in the upper zone of the tank during the course of six 10-min exposures to the same novel tank. (**E**) Plot of the average time it took for a fish to first enter the upper zone of the tank during the course of six 10-min exposures to the same novel tank. For all data, N=4 groups per genotype, 23 total fish per genotype. Error bars = SEM. For figures D-F, videos that could not be automatically tracked were omitted (for WT, 6/138, for *klc4<sup>uw314</sup>*, 4/138).

The online version of this article includes the following source data and figure supplement(s) for figure 12:

**Source data 1.** The percentage of fish per group that froze during the first 3 min of exposure 1 of the habituation series.

**Source data 2.** The percentage of fish per group that froze for each exposure in the habituation series.

**Source data 3.** The total time spent frozen by each individual fish throughout the experimental time period for each exposure in the habituation series.

**Source data 4.** The mobile and total swim velocities for each individual fish throughout the experimental time period for each exposure in the habituation series.

**Source data 5.** The percentage of fish per group that entered the upper zone of the tank at least once during each exposure in the habituation series.

**Source data 6.** The time in seconds it took each individual fish to initially cross over into the upper zone of the tank for each exposure in the habituation series.

**Figure supplement 1.** Additional data for novel tank habituation experiments.

exploration of the upper zone and reduced freezing behavior. Fish that do not habituate to the novel environment or that worsen over multiple exposures are thought to demonstrate a depression-like phenotype (*Ziv et al., 2013*). Behaviors driven by increased anxiety are expected to lessen with repeat exposures. We tested the ability of *klc4*[uw314] mutants to habituate to a novel environment after multiple exposures. In addition, because KLC4 and other kinesin-1 subunits have been implicated in neurodegenerative diseases, we sought to determine whether the anxiety-like behavior worsened with age. We challenged 1.25–1.5 year old *klc4*[uw314] mutants and their wild type cousins by repeated exposure to a novel tank for 10 min, twice a week for 3 weeks in succession. The first exposure allowed us to test the effects of age. We found that increased age improved freezing response of *klc4*[uw314] mutants, with only 29.2% (8 out of a total of 23) of individuals tested freezing for more than 3 s during first 3 min of exposure to the novel tank (*Figure 12A*). Aged wild-type animals worsened in performance, with 13.3% (3 out of 23) undergoing a bout of freezing behavior during first exposure. These numbers remained similar when the entire 10 min was analyzed (wt = 16.7%, mut = 29.2%, *Figure 12B*, *Figure 12—figure supplement 1A*). Mutant fish spent a longer period of time frozen than wild-type fish and underwent more total bouts of freezing per fish (*Figure 12C*, *Figure 12—figure supplement 1B*). Upon subsequent exposures to the same tank, *klc4*[uw314] mutant freezing behavior worsened (*Figure 12B*, *Figure 12—figure supplement 1A, B*), with 50% of *klc4*[uw314] mutants freezing by the sixth exposure, indicating that mutants fail to habituate freezing behavior. Mutants consistently froze for longer periods of time than wild type animals throughout all exposures (*Figure 12B and C*). Wild type animals performed approximately equally through subsequent exposures, undergoing less frequent freezing bouts and spending less time frozen than mutants (*Figure 12B and C*, *Figure 12— figure supplement 1A, B*). Wild-type fish also swam faster than *klc4*[uw314] mutants and showed similar swim velocities when their total swim velocity (including pauses) was compared to their mobile swim velocity (with pauses in swimming subtracted) (*Figure 12D*). *klc4*[uw314] swim mobile swim velocity was faster than their total swim velocity, reflecting that they spent a greater amount of time immobile (*Figure 12D*).

Interestingly, both wild type and *klc4*[uw314] mutants did show habituation as measured by the percentage of time spent in the upper zone of the tank (*Figure 12E*). During the first exposure, mutant animals only spent 8.7% of the ten minute experiment window in the upper zone and took on average 170.2 s before exploring the upper zone. Wild-type animals explored the upper zone after an average of 80.0 s and spent 17.6% of the first exposure period in the upper zone (*Figure 12E and F*). At the time of the sixth exposure, however, both wild type and *klc4*[uw314] mutants increased their time spent in the upper zone, and mutants entered the upper zone faster than in their first exposure (*Figure 12E and F*). Wild type animals took on average 79.7 s to explore the upper zone and spent an average of 25.4% of the experiment period in the upper zone. *klc4*[uw314] mutants performed similarly, spending 23.1% of the experiment period in the upper zone, but taking on average 129.0 s to explore the upper zone (*Figure 12E and F*). Thus, the failure of *klc4*[uw314] mutants to habituate freezing behavior is specific rather than reflecting a general deficit in habituation.

## Discussion

The association of KLC4 with human disease demonstrates its critical importance, yet its cellular and developmental functions are poorly understood. In this study, we used automated analysis of sensory axon arbors and live, in vivo imaging of axon behavior and intracellular MT dynamics to define new roles for KLC4 in shaping neuronal morphology. We found that KLC4 is required for sensory axon branching and for repulsion between peripheral axons, behaviors which are important for proper innervation and tiling of the sensory field. *Klc4* mutant neurons do not have a general growth defect, in that their axons grow faster than wild type axons. In addition, *klc4* mutants are adult viable and show no signs of early neurodegeneration. These results suggest that KLC4 has a specific function in patterning developing neurons rather than simply being required for general neuronal health. Our finding that the mutant larvae are hypersensitive to touch stimuli suggests that the altered sensory arborization pattern affects sensory circuits. At least two axon defects could contribute to the increased response. Fasciculation of peripheral axons from neighboring neurons might lead to an increase in the number of neurons activated by a localized touch stimulus. Also, the aberrant midline crossing of axons in mutants could cause inappropriate activation of contralateral neurons. Behavioral analysis also showed that *klc4* mutant adults exhibit increased anxiety-like behavior, indicating roles for KLC4

in adult circuit function and identifying KLC4 as a novel gene associated with anxiety behavior. This work sheds light on potential mechanisms of human HSPs and provides a new animal model uniquely suited to further investigation of the cell biological and circuit mechanisms underlying disease.

Precise regulation of axon branching is critical for neurons to develop diverse and complex architecture. We identified KLC4 as a novel regulator of axon branching and show that it is required for stabilization of nascent branches. Axon branching requires extensive cytoskeletal rearrangements (*Kalil and Dent, 2014*; *Gallo, 2011*; *Winkle et al., 2016*). Branches begin as actin-based filopodial protrusions, which must be invaded by MTs to mature into branches. MT severing by proteins such as katanin and spastin is required to facilitate invasion (*Qiang et al., 2010*; *Yu et al., 1994*; *Yu et al., 2008*) and branch points are hot spots of increased MT dynamics (*Dent and Kalil, 2001*). Local unbundling of MTs in the axon shaft occurs at sites of interstitial branch formation (*Kalil and Dent, 2014*; *Ketschek et al., 2015*). Kinesin-1 can bind MTs via the KHC C-terminal tail and crosslink or bundle MTs (*Seeger and Rice, 2010*; *Navone et al., 1992*). Binding of KLC to KHC releases the autoinhibition of KHC and can tip the balance between kinesin-1's cargo transport activities and the ability of the KHC tail region to bind MTs (*Wong and Rice, 2010*; *Cai et al., 2007*; *Chiba et al., 2021*). Our finding that MT dynamics are affected in *klc4* mutants is consistent with a model in which KLC4 promotes the localized MT dynamics or severing that allows MTs to invade new branches, either by transporting/localizing MT regulators or by facilitating MT unbundling. The increase in MT velocity could suggest an increase in MT sliding or altered regulation or localization of polymerization-promoting MAPs such as CRMP2 (*Nakamura et al., 2020*). CRMP2 has been reported to influence MT polymerization and stabilization and has been found in complex with KLC4, supporting its potential role in KLC4-mediated effects on MTs (*Liz et al., 2014*; *Koh et al., 2021*). The model of KLC4 as a mediator of microtubule dynamics is also supported by the similarities in phenotype between *klc4* mutants and kinesore treatment. Kinesore blocks binding between KLCs and cargo adaptors, and also releases KLC autoinhibition and drives KHC toward increased MT bundling function (*Randall et al., 2017*). MTs that invade nascent branches in *klc4* mutants have less acetylation, suggesting KLC4 may influence post-translational modifications of MTs and thereby affect stabilization of new branches.

In addition to cytoskeletal remodeling, regulated organelle/cargo transport and membrane remodeling are important for axon branching (*Winkle et al., 2016*), although little is known about how individual regulators of trafficking contribute to branching patterns. We previously found that a properly functioning endosomal trafficking system is necessary for RB axon branching (*Ponomareva et al., 2014*; *Ponomareva et al., 2016*). Moreover, we showed that the kinesin adaptor Clstn1 is required for branching of peripheral RB axons and that it functions in part by trafficking endosomal vesicles to branch points (*Ponomareva et al., 2014*). KLC4 loss has a relatively minor effect on Rab5 endosome transport, which is not likely to fully explain the branch stabilization phenotype. KLC4 could potentially influence transport or localization of other necessary organelle cargos, such as mitochondria, which must be localized to branch points to support branch growth (*Courchet et al., 2013*; *Spillane et al., 2013*; *Tao et al., 2014*; *Wong et al., 2017*), or organelles that support mitochondria, such as ER, which is a critical regulator of mitochondrial function (*Csordás et al., 2018*; *Rowland and Voeltz, 2012*; *Krols et al., 2016*). An important future research goal is to determine whether KLC4 localizes specific organelles to branch points.

To maximize coverage of an innervation field, the axonal or dendritic arbors of neuronal populations that share a similar function must repel one another on contact to 'tile' the field (*Grueber and Sagasti, 2010*; *Spead and Poulain, 2020*). Our results show that KLC4 is an important player in repulsion between sensory axons, and to our knowledge KLC4 is the first motor protein shown to be involved in tiling. Several cell surface or secreted proteins that mediate dendritic tiling have been identified, including protocadherins (*Kostadinov and Sanes, 2015*; *Lefebvre et al., 2012*), DSCAM (*Matthews et al., 2007*; *Soba et al., 2007*), netrin (*Smith et al., 2012*), and slit/robo signaling (*Gibson et al., 2014*). Less is known about the mechanisms of axon tiling, although similar molecules appear to be involved. For example, in *Drosophila*, DSCAM mediates sister branch separation in mushroom body axons (*Wang et al., 2002*), and Dscam2 mediates repulsion between axons in the visual system (*Millard et al., 2007*). An atypical cadherin, Flamingo, is required for spacing of photoreceptor axon arbors (*Senti et al., 2003*), and functions together with the transmembrane protein Golden goal (*Tomasi et al., 2008*; *Hakeda-Suzuki et al., 2011*). More recent studies have shown that the protocadherin PcdhaC2 mediates repulsion between axons of serotonergic neurons in the mammalian brain

(*Katori et al., 2017*; *Chen et al., 2017*). Thus, the field has seen multiple advances in knowledge of the membrane proteins that mediate repulsion. However, the mechanisms that regulate their localization to the correct cellular compartments are unknown. Our work can drive new exploration into such mechanisms. For example, it will be interesting to determine whether KLC4 mediates transport of contact repulsion molecules and their specific targeting to peripheral versus central sensory axons.

The formation of polarized neuronal structure is a critical component of proper neural circuit development. An important early step in neuronal polarity development is determination of the location of axon emergence from the cell body. The ectopic apical axons we see in *klc4* mutants suggest that KLC4 also has roles in the polarity of axon initiation location. RB neuron axons normally emerge from basal edge of cell body (*Andersen et al., 2011*), and their emergence correlates with centrosome movement to the basal region of the cell (*Andersen and Halloran, 2012*). We previously found that *laminin-1* mutants also have ectopic apical axons in RB neurons and these ectopic axons correlate with apically displaced centrosomes (*Andersen and Halloran, 2012*). Laminin has been shown to influence the location of axon emergence from the cell body in zebrafish RGCs (*Randlett et al., 2011*), and can direct the positioning of the centrosome in neurons in vitro and in vivo (*Randlett et al., 2011*; *Gupta et al., 2010*). KLC4 may play a role in coordinating the response to extracellular signals such as laminin that help define the site of axon formation.

Overall, the developmental phenotypes seen in *klc4[uw314]* mutants are striking in that they appear to reflect problems with specific patterning of neuronal structure rather than problems in bulk cargo transport. Neurons appear healthy in the mutants, show no signs of axon degeneration during development, and in fact axons grow faster in mutants than in wild type embryos. Moreover, the *klc4[uw314]* mutants are adult viable. Interestingly, *klc1* knockout mice are also adult viable and show specific defects in transport of cannabinoid receptors in developing axons (*Saez et al., 2020*). Thus our findings may reflect a general theme, that the function of the KLC subunits of the kinesin-1 motor are to provide specificity to the process of localizing cargos to particular cell regions. Our results also give insight into mechanisms of the human disease caused by *KLC4* mutation, which may be caused by defects in developmental patterning. It will be important to fully define the unique cellular functions of individual KLCs in order to better understand mechanisms of disease.

Adult circuit function can be affected by both defects in circuit development and/or later defects in the function of neurons within circuits. Our finding that *klc4[uw314]* mutant adults show freezing and reduced exploration in the novel tank assay implicates KLC4 in the circuits that regulate stress response and that are affected in anxiety and depression disorders. The hypothalamic-pituitary-adrenal (HPA) axis is an important system that mediates stress responses, and zebrafish mutants with disruptions to the HPA also exhibit anxiety behavior in the novel tank test. Mutation of zebrafish glucocorticoid receptor, a target of the stress hormone cortisol, results in increased freezing and decreased exploratory behavior. Like *klc4[uw314]* mutants, glucocorticoid receptor mutants fail to habituate freezing behavior after repeated exposures to the novel tank (*Ziv et al., 2013*). However, unlike *klc4[uw314]* mutants, glucocorticoid receptor mutants do not habituate exploratory behavior over multiple exposures. This suggests that freezing behavior and exploratory behavior are controlled by different mechanisms. Interestingly, zebrafish with mutation in a chemokine gene expressed in multiple brain regions, *sam2*, freeze during the novel tank test, but show normal tank exploration (*Choi et al., 2018*), providing additional evidence that these two behaviors can be uncoupled. The *sam2* mutants have disruptions in corticotropin releasing hormone neurotransmission, linking the mutation to HPA disruption, but not excluding contribution from other circuits. Our *klc4[uw314]* mutants provide another tool to further investigate the circuit changes underlying anxiety disorders.

Whether developmental defects in RB neurons contribute to adult circuit function is unknown. RB neurons are traditionally thought to be a transient cell population that undergoes programmed cell death during larval stages. However recent evidence has shown that 40% of RBs are maintained for at least the first 2 weeks of development, suggesting they may be retained (*Williams and Ribera, 2020*). Moreover, RB neurons have been shown to play significant roles in regulating larval behaviors (*Knafo et al., 2017*). Hypersensitivity to touch is a common trait of autism spectrum disorder (ASD), although the relationship of somatosensory neurons to states like anxiety, depression, and hyperstimulation is not well defined. Interestingly, deletion of the ASD risk genes Mecp2 and Gabrb3 in mouse primary somatosensory neurons during development led to a reduction in exploratory behavior in the open-field test, while deletion of these genes only during adulthood caused no such impairment or anxiety

behavior (*Orefice et al., 2016*). In addition to sensory neurons, *klc4* is expressed in many cell types in the developing nervous system and thus likely has important roles in development and function of other neurons, which may include neurons that are directly involved in circuits mediating adult behavior. A future research goal is to test the functions of KLC4 in neurons of stress response circuits, such as the HPA axis.

# Materials and methods

## Key resources table

| Reagent type (species) or resource | Designation | Source or reference | Identifiers | Additional information |
|---|---|---|---|---|
| Gene (*Danio rerio*) | klc4 | Ensembl | ENSDARG00000086985 | |
| Strain, strain background (*Danio rerio*) | AB | ZIRC | ZL1 | Wildtype Strain of Zebrafish |
| Strain, strain background (*Danio rerio*) | TL | ZIRC | ZL86 | Wildtype Strain of Zebrafish |
| Genetic reagent (*Danio rerio*) | Tg(–3.1ngn1:gfp-caax) | *Andersen et al., 2011* | Tg(ngn:gfp-caax) | Transgenic line labelling sensory axons, AB/TL background |
| Recombinant DNA reagent | 3.1ngn1:trfp-caax | *Andersen et al., 2011* | Tg(ngn:trfp-caax) | Plasmid for labelling sensory axons |
| Recombinant DNA reagent | 3.1ngn1-mcherry-caax | *Andersen et al., 2011* | Tg(ngn:mcherry-caax) | Plasmid for labelling sensory axons |
| Recombinant DNA reagent | 3.1ngn1:eb3-caax | *Stepanova et al., 2003*; *Ponomareva et al., 2014* | Tg(ngn:eb3-gfp) | Plasmid for labelling microtubule plus ends |
| Recombinant DNA reagent | 3.1ngn1:GFP-rab5c | *Clark et al., 2011*; *Ponomareva et al., 2014* | Tg(ngn:GFP-rab5c) | Plasmid for labelling Rab5c vesicles |
| Recombinant DNA reagent | 3.1ngn1:PA-gfp-Rab5c | *Ponomareva et al., 2014* | Tg(ngn:PA-gfp-Rab5c) | Plasmid for labelling Rab5c vesicles |
| Sequence-based reagent | klc4 in situ probe | This paper | | Sequences listed in methods |
| Sequence-based reagent | klc4 long isoform specific in situ probe | This paper | | Sequences listed in methods |
| Sequence-based reagent | short isoform RT primers | This paper | | Sequences listed in methods |
| Sequence-based reagent | long isoform RT primers | This paper | | Sequences listed in methods |
| Sequence-based reagent | klc4 gRNA | This paper | | Sequence listed in methods |
| Peptide, recombinant protein | Cas9 protein with NLS | PNA Bio | NC0789474 | |
| Chemical compound, drug | Trizol | Thermo-Fisher | 15596026 | For RNA extraction |
| Chemical compound, drug | Kinesore | Chembridge Hit2Lead library | #6233307 | Kinesin1 targeting small molecule |
| Commercial assay or kit | Superscript III | Invitrogen | 18080093 | |
| Commercial assay or kit | DIG RNA labelling kit | Roche | 11175025910 | |
| Chemical compound, drug | Oligo dT | Invitrogen | 18418020 | |
| Chemical compound, drug | Random Hexamers | Invitrogen | N8080127 | |
| Antibody | anti-HNK1 (mouse monoclonal) | ZIRC | ZN-12 | IF(1:250) |
| Antibody | anti-acetylated tubulin (Mouse monoclonal) | Sigma | T6793 | IF(1:1000) |
| Antibody | anti-mouse IgG AlexaFluor 488 (goat polyclonal) | Invitrogen | A-21121 | IF(1:1000) |
| Antibody | Goat anti-mouse IgG2b AlexaFluor 647 (goat polyclonal) | Invitrogen | A-21141 | IF(1:1000) |
| Software, algorithm | CRISPRscan | *Moreno-Mateos et al., 2015* | | https://www.crisprscan.org/ |
| Software, algorithm | CRISP-ID | *Dehairs et al., 2016* | | http://crispid.gbiomed.kuleuven.be/ |
| Software, algorithm | ToxTrac | *Rodriguez et al., 2018* | | Animal Tracking Freeware |
| Software, algorithm | OrientationJ | *Rezakhaniha et al., 2011* | | Used to quantify orientation of axons |

*Continued on next page*

*Continued*

| Reagent type (species) or resource | Designation | Source or reference | Identifiers | Additional information |
|---|---|---|---|---|
| Software, algorithm | Graphpad Prism | Graphpad | | Graph making software |
| Software, algorithm | Chemotaxis Tool | Ibidi | | https://ibidi.com/chemotaxis-analysis/171-chemotaxis-and-migration-tool.html |

## Animals

Adult zebrafish (*Danio rerio*) were kept in a 14/10 hr light/dark cycle. Embryos were maintained at 28.5 °C and staged as hours postfertilization (hpf) as described previously (*Kimmel et al., 1995*). Wild type AB strain or transgenic Tg(−3.1ngn1:GFP-CAAX) (*Andersen et al., 2011*) (AB/TL background) embryos were used for all experiments. In experiments using the *Tg(−3.1ngn1:GFP-CAAX)* line as a background, wild type is used to refer to *Tg(−3.1ngn1:GFP-CAAX)*, *klc4*$^{+/+}$ animals. All animals in these studies were handled in accordance with the National Institutes of Health Guide for the care and use of laboratory animals, and the University of Wisconsin Institutional Animal Care and Use Committee (IACUC). These studies were approved by the University of Wisconsin IACUC (protocols L005692 and L005704).

## CRISPR/Cas9 mutagenesis

A guide RNA targeting zebrafish *klc4* (5′-GAGCGTGGCACAGCTGGAGGAGG-3′) was designed by using CRISPRScan (crisprscan.org) and synthesized as previously described (*Shah et al., 2015*; *Moreno-Mateos et al., 2015*). To generate *klc4*$^{uw314}$ mutants, embryos at the very early one-cell stage were injected with 374.5 ng/µl of the guide RNA targeted to zebrafish *klc4* complexed with 800 ng/µl of Cas9 protein with NLS (PNA Bio). Injection volume was approximately 1 nL. F0 adult fish were pair-mated to determine individuals carrying germ line mutations. A subset of individuals with germ line mutations were outcrossed to wild type AB fish. F1 fish were raised and genotyped by PCR and sequencing, using CRISP-ID (http://crispid.gbiomed.kuleuven.be/) to assist in sequence unmixing (*Dehairs et al., 2016*). The *klc4*$^{uw314}$ allele was found to have a 25 bp deletion. Fish heterozygous for *klc4*$^{uw314}$ were outcrossed to wild type AB fish until the F3 generation. F3 *klc4*$^{uw314}$ heterozygotes were in-crossed to generate F4 *klc4*$^{uw314}$ homozygotes. All homozygous mutants in this study were F5-F6 generation mutants from homozygous parents, unless otherwise specified. Adults and embryos described as "wild type cousins" of *klc4*$^{uw314}$ mutants were generated by crossing F4 *klc4*$^{+/+}$ siblings from the F3 *klc4*$^{uw314/+}$ in-cross.

## RT-PCR

RNA was extracted from zebrafish embryos at specified ages with Trizol (Thermo-Fisher), followed by chloroform extraction and isopropanol precipitation. cDNA was generated from embryo-derived RNA using the Superscript III reverse transcriptase kit and a 50:50 mixture of oligo dT and random hexamer primers (Invitrogen). We designed PCR primers to amplify specific *klc4* isoforms. A reverse primer unique to the short isoform's 3′ UTR was designed for the *klc4* short isoform, with the forward primer in exon 2 (Fwd: 5′-GCGCAAATCAGTGGAGATGATAGAGC-3′, Rev: 5′-GCATAGTGGCGCGGTAGGTA-3′). A set of primers for exons exclusive to the long isoform of *klc4* (spanning exons 7 through 12) was designed for the long isoform (Fwd: 5′- CAGAGAGAAGGTGCTGGG –3′, Rev: 5′-GGTGGTGTTCACTGTTGG-3′). PCR was then run to test for presence of the isoforms in cDNA from each embryonic age.

## In situ hybridization

A digoxigenin-labeled riboprobe complementary to a region spanning exons 2–3 of the *klc4* mRNA, which are contained in both long and short isoform mRNAs, was synthesized by in vitro transcription using T7 RNA polymerase with a DIG RNA labeling kit (Roche). The template for transcription was made using PCR on zebrafish cDNA with the following primers: Fwd: 5′-GCGCAAATCAGTGGAGATGATAGAGC-3′, Rev: 5′-TAATACGACTCACTATAGCTCCAGATGTTTCTTCTCCTCC-3′. A second probe recognizing a region in exons 8–11, found only in the long isoform, was synthesized using

PCR on zebrafish cDNA with the following primers: Fwd: 5'-CAG AGA GAA GGT GCT GGG-3', Rev: 5'- TAATACGACTCACTATAG GGTGGTGTTCACTGTTGG-3'. Whole-mount in situ hybridization was performed as previously described (*Halloran et al., 1999*).

## Immunohistochemistry

Embryos were fixed overnight at room temperature in 4% paraformaldehyde in 0.1 M PO$_4$ buffer, then rinsed with PBS/0.1% Tween-20 (PBST) and blocked with incubation buffer (IB: 2 mg/mL BSA, 1% Sheep Serum, 1% DMSO, 0.5% Triton X-100 in PBS) for at least 1 hr. Embryos were incubated overnight at 4 °C with monoclonal anti-HNK1 antibody in IB (ZN-12, 1:250; Zebrafish International Resource Center), rinsed in PBST, then incubated with anti-mouse AlexaFluor 488 secondary antibody (1:1000; Invitrogen) for at least 2 hr at room temperature or overnight at 4 °C. For double labeling with anti-acetylated tubulin (T6793, 1:1000, Sigma), the IgG subtype-specific secondary antibodies anti-mouse IgG1 AlexaFluor 488 and anti-mouse IgG2b AlexaFluor 647 (1:1000, Invitrogen) were used.

## Microinjection and Mosaic Expression

DNA expression constructs were made using the Multisite Gateway Cloning System (Invitrogen) into Tol2 vectors (*Kwan et al., 2007*). Zebrafish embryos at the one-cell stage were injected with ~10 pg –*3.1ngn1:EB3-GFP* DNA (*Lee et al., 2017*) to visualize microtubule dynamics. For experiments to image individual neuron morphology, embryos were injected with ~12 pg *3.1ngn1:mCherry-CAAX* DNA (*Andersen et al., 2011*).

## Kinesore treatment

Kinesore was acquired from the Hit-2-Lead library (Chembridge, compound #6233307) and suspended in DMSO for a stock concentration of 100 mM. Embryos treated with Kinesore were dechorionated at 17 hpf and placed in E3 containing 150 μM Kinesore and 2% DMSO. Controls were treated with 2% DMSO in E3 for the same time period.

## Live Imaging

Light sheet imaging was performed on a custom multi-view light sheet microscope as described previously (*Daetwyler et al., 2019*). The detection objective was either an Olympus UMPLFLN 10 x W (NA 0.3) or a Nikon CFI75 Apochromat 25xC W (NA 1.1) with a 400 mm tube lens (TL400-A) for 50 x total magnification. Briefly, the Tg(*3.1ngn1:GFP-CAAX*) wild type or mutant embryos were mounted in an FEP tube (0.8 mm inner diameter, 1.2 outer diameter, Bola) suspended in 0.02% tricaine in E3 with an agarose plug to prevent leakage.

For live confocal imaging, embryos were anesthetized in 0.02% tricaine and mounted in 1% low melting agarose in 10 mM HEPES E3 medium as described previously (*Andersen et al., 2011*). Axon outgrowth was imaged in Tg(*3.1ngn1:GFP-CAAX*) embryos with either an Olympus FV1000 laser-scanning confocal microscope using a UplanSApo 20 x Oil objective (NA 0.85) or an Opterra Swept-Field confocal (Bruker Fluorescence Microscopy) with a Nikon S Fluor 40 x Oil immersion objective (NA 1.30). Imaging was started between 18 and 19 hpf, when peripheral RB axons are extending out of the spinal cord. Optical sections (1–1.25 μm) covering a 30–36 μm Z depth were captured every 60–90 s for 2–4 hr.

High speed imaging of EB3-GFP comets was performed with the Opterra Swept-Field confocal microscope equipped with a Nikon Plan Apo VC 100 x oil immersion objective (NA 1.40). Embryos were imaged at stages between 18 and 26 hpf, during peripheral axon outgrowth. EB3-GFP comets in nascent branches were imaged at stages between 22 and 25 hpf with a Nikon Plan Apo 60 x oil immersion objective (NA 1.40). Z-stacks of 5–30 1 μm optical sections were captured at 2–5 s time intervals, for total durations between 7 and 10 min. High speed imaging of Rab5 vesicle dynamics was done with the Opterra Swept-Field confocal and the Nikon Plan Apo 100 x oil objective (NA 1.40). Embryos were imaged between 22 and 26 hpf. Z-stacks of 5–30 μm optical sections were captured at 2–5 s time intervals for a total duration of 7–10 min. Photoactivation and imaging of PA-GFP-Rab5 were performed on the Olympus FV1000 confocal using a 60 X UplanSApo oil immersion objective (NA 1.35). A region of interest encompassing the cell body was activated with the 405 nm laser during the scan of one Z-plane in the cell body. Z-stacks of 1 μm optical sections were captured every 30–170 s for durations ranging from 26 to 34 minutes (average 29.22 minutes, median 28.84 minutes).

## Axon outgrowth analysis

To measure the speed of axon outgrowth, the area immediately behind the growth cone was measured using the Manual Tracking plugin in FIJI (Fabrice Cordelières). The generated data was loaded into the Chemotaxis and Migration Tool (Ibidi, https://ibidi.com/chemotaxis-analysis/171-chemotaxis-and-migration-tool.html) to generate velocity measurements and rose plot histograms. Rose plots were re-plotted in MATLAB (MathWorks) using secplot (Matthew Foster, https://www.mathworks.com/matlabcentral/fileexchange/14174-secplot).

## Branch analysis

Custom macros for FIJI (*Schindelin et al., 2012*) were written to automate branch analysis. Briefly, stacks were processed to be maximum intensity projections and oriented with anterior to the left. A rectangular ROI was selected over the posterior region of the yolk tube extension such that the top of the selection was aligned with the bottom of the central RB axons. A tubeness filter was applied to reduce non-axonal background. A histogram of axon directionality was generated for the ROI using OrientationJ (*Rezakhaniha et al., 2011*). To measure differences in axon directional distribution, we computed the area under the curve (AUC) for the regions to be compared. We then performed a t-test using the resulting mean and standard error to determine significance. To measure axon branch density, eight ROIs measuring 3 μm in height were generated at regularly spaced intervals from dorsal to ventral (*Figure 2—figure supplement 1*). These selections were individually thresholded, binarized, and the number of objects contained in each ROI was determined using the Analyze Particles function. To analyze acetylated tubulin signal in nascent branches, terminal branches up to 10 μm in length were scored based on the presence or absence of visible acetylated tubulin signal. Total proportion of nascent branches with signal was compared between wild type and mutant embryos.

## Kymography

Hyperstacks of EB3-GFP or GFP-Rab5c images were assembled using FIJI (*Schindelin et al., 2012*) and segments of axons were traced for kymography using the segmented line tool. For peripheral axons, a segment was defined as an area of axon between two branch nodes, or between the first visible portion of axon and the first node of branching. Growth cones and filopodia were excluded from analysis. For analysis of EB3-GFP in nascent branches, only branches < 10 μm long were analyzed. For unbranched central axons, a segment was defined as the entire visible length of either the ascending or descending central axon within each field of view. Kymographs were made using the Multi Kymograph plugin (developed by J. Rietdorf and A. Seitz, European Molecular Biology Laboratory, 2004). Comet or vesicle parameters were measured from the resulting kymographs. To measure Rab5 accumulation at branch points, movies were divided into 4 time periods (t1-t4), each containing 7–8 min. Maximum intensity projections were made for t1-t4 and the fluorescent signal at branch points was measured for each maximum intensity projection. Background signal from the branch point prior to activation was subtracted from measured t1-t4 values, and then all values were normalized to the signal at the t1 period. Comparisons between experimental conditions were made using the Mann-Whitney test for non-parametric data.

## Adult Body Size Measurements

Zebrafish were raised for this experiment by in-crossing wildtype or *klc4*^*uw314*^ homozygous siblings. Larva were placed in 3 L tanks side-by-side and subjected to the same light cycle and feeding schedule during rearing. At 7 weeks and 9 weeks, fish were briefly anesthetized with 0.02% tricaine, placed on white waterproof surface, measured with a ruler, and photographed.

## Larval touch assay

To assess responsiveness to touch, 3 dpf zebrafish larva were individually placed in a 6 cm dish and allowed to acclimate for 2 min. A tail touch was administered to the end of the tail using a gel loading tip attached to a long applicator stick. The tail touch was given from a side angle to avoid looming stimulus. A high-speed camera was used to acquire 50 frames per second for a total of 1050 frames (21 s). The response time of each larva was measured from the first movement after a touch to the cessation of movement for more than 15 frames (inter-bout pauses only lasted ~7–8 frames). Some

wild type and *klc4* mutant larva did not respond to an initial touch (10 wild type, 14 mutant), and these samples were excluded from analysis.

## Feeding experiment

Feeding experiments were conducted as in *Howe et al., 2018*. Briefly, adult zebrafish were isolated in individual 1 L tanks without food for 48 hr, then 50 larval zebrafish were added to each tank. The fish were allowed to feed on the larvae for one hour, then removed from the tank to allow counting of the remaining larvae. All tested animals were of AB background.

## Novel tank diving test

The evening before an experiment, fish were moved to the behavior experiment room and allowed to acclimate overnight. Exposure to novel tank was carried out in the morning, and the order of which genotype received exposure first was randomized. The novel tank had 3 opaque sides to block potential views of other fish and one clear side for imaging. The water in the novel tank was changed between groups. Fish were netted from their 3 L home tank, released into the novel tank, and allowed to explore for 3 min before being removed and placed into a recovery tank. All tested animals were of AB background.

## Habituation experiment

Two weeks prior to an experiment, adults were sorted into separate 3 L tanks with 5–6 individuals of the same genotype in each. Experiment was conducted in a similar manner as described above, but fish were allowed to explore for 10 min before removal. Novel tank exposure was repeated on the same groups twice a week for 3 weeks. Exposures were conducted a minimum of two days apart. The experiment was repeated for another 3 weeks with an additional two groups of 6 fish per genotype. All tested animals were of AB background. When possible, groups were divided into similar ratios of male and female fish for a total of 11 females and 12 males within each genotype. The group compositions were as follows: **wild type**: group 1 (3 females [F], 2 males [M]) group 2, (3 F, 3 M), group 3 (3 F, 3 M), group 4 (2 F, 4 M), ***klc4^uw314^***: group 1 (2 F, 3 M), group 2, (3 F, 3 M), group 3 (3 F, 3 M), group 4 (3 F, 3 M).

## Automated tracking of adults

For the habituation experiment, automated tracking of zebrafish through ToxTrac (*Rodriguez et al., 2018*) was used to quantify swim velocity and percent of time spent in the upper zone. When necessary for accurate tracking, background subtraction was performed on movies prior to using ToxTrac. This was accomplished by subtracting the first frame of each movie (showing only the empty tank) from subsequent frames in the movie via FIJI (*Schindelin et al., 2012*). Operations were performed using Clij2 where possible to speed up processing time (*Haase et al., 2019*).

## Statistics

Statistical analyses and graphs were generated with Graphpad Prism. See figure legends for details on sample sizes, specific statistical tests used, error bars and p values. Data were checked for normality using Anderson-Darling test, D'Agostino & Pearson test, and Shapiro-Wilk test before deciding on a statistical test. Non-normal data were tested for significance using a Mann-Whitney test. In some cases, the N values were too low to perform a statistical test (*Figure 6E*, *Figure 11E and H*, *Figure 12A*). In *Figure 6E*, the N for wild type is low because fasciculation events are very rare in wild type and obtaining a value for fasciculation length is contingent on the event occurring. For the behavioral data shown in *Figure 11E and H* and *Figure 12A*, multiple animals were tested in each trial, but N is the number of individual behavioral trial days (N=3, which is too low to test for normality). In all these cases, individual data points are shown for comparison. Data tested using the Mann-Whitney test were independent of each other and comprised ordinal or continuous numbers.

## Acknowledgements

The authors thank the members of the Huisken lab for technical expertise and intellectual contributions, particularly Kurt Weiss, Joe Li, and Alyssa Graves. We are grateful to Katie Drerup and Bill Bement for critical reading and feedback on this manuscript, which greatly improved the final work.

We also give our heartfelt thanks to the current and former members of the Halloran lab for their support of and contributions to this work, especially Conlin Bass for valuable work during the initial characterization of *klc4* expression pattern, and Daniel North for fish care and maintenance. This work was supported by funds to Mary Halloran from the National Institutes of Health (R01 NS086934, R21 NS116326) and funds to Elizabeth Haynes from the National Institutes of Health (F32 NS098689) and the Morgridge Institute for Research. The Morgridge Institute for Research provided support for the Huisken lab and the light sheet microscope. The Center for Quantitative Cell Imaging provided support for the Opterra Confocal.

## Additional information

### Competing interests
Kevin W Eliceiri: is a consultant for Bruker, the manufacturer of the Opterra swept field confocal used in this work. The other authors declare that no competing interests exist.

### Funding

| Funder | Grant reference number | Author |
| --- | --- | --- |
| National Institutes of Health | R01 NS086934 | Mary C Halloran |
| National Institutes of Health | R21 NS116326 | Mary C Halloran |
| National Institutes of Health | F32 NS098689 | Elizabeth M Haynes |

The funders had no role in study design, data collection and interpretation, or the decision to submit the work for publication.

### Author contributions
Elizabeth M Haynes, Conceptualization, Software, Formal analysis, Supervision, Funding acquisition, Investigation, Visualization, Writing – original draft, Writing – review and editing; Korri H Burnett, Formal analysis, Investigation, Writing – review and editing; Jiaye He, Conceptualization, Resources, Software, Supervision, Investigation, Methodology, Writing – review and editing; Marcel W Jean-Pierre, Conceptualization, Software, Formal analysis, Supervision, Investigation, Visualization, Writing – original draft, Writing – review and editing; Martin Jarzyna, Conceptualization, Software, Investigation; Kevin W Eliceiri, Resources, Supervision, Investigation, Methodology, Writing – review and editing; Jan Huisken, Resources, Methodology, Writing – review and editing; Mary C Halloran, Conceptualization, Supervision, Funding acquisition, Investigation, Writing – original draft, Writing – review and editing

### Author ORCIDs
Elizabeth M Haynes http://orcid.org/0000-0002-5294-018X
Kevin W Eliceiri http://orcid.org/0000-0001-8678-670X
Jan Huisken http://orcid.org/0000-0001-7250-3756
Mary C Halloran http://orcid.org/0000-0001-6086-5928

### Ethics
This study was performed in accordance with the recommendations in the Guide for the Care and Use of Laboratory Animals of the National Institutes of Health. Animals were handled according to approved institutional animal care and use committee protocols of the University of Wisconsin (protocols L005692 and L005704).

### Decision letter and Author response
Decision letter https://doi.org/10.7554/eLife.74270.sa1
Author response https://doi.org/10.7554/eLife.74270.sa2

## Additional files

**Supplementary files**
• Transparent reporting form

**Data availability**
All data generated or analyzed in this study are included in the manuscript and supporting files. Source Data files have been provided for Figures 1-12.

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
