## [Editor Report]

Using zebrafish as an in vivo model, this study reveals for the first time that mutations in the kinesin light chain gene klc4, which are known to cause a form of early onset hereditary spastic paraplegia in human, affect specific aspects of neuronal development and nervous system functions. High-resolution movies of developing sensory neurons in vivo and behavioral assays support the key findings that the motor subunit Klc4 plays essential functions in the control of neuronal morphogenesis and compartmentalization as well as associated behaviors.

---

## [Decision Letter]

**Decision letter after peer review:**

Thank you for submitting your article "KLC4 shapes axon arbors during development and mediates adult behavior" for consideration by *eLife*. Your article has been reviewed by 3 peer reviewers, including Fabienne E. Poulain as Reviewing Editor and Reviewer #1, and the evaluation has been overseen by Didier Stainier as the Senior Editor.

After discussing their reviews with one another, and as you will see from the individual reviews below, all three reviewers agree that your study describes novel and interesting functions of Klc4 in sensory axon development and has therefore a potential high impact. However, reviewers share four major concerns that need to be fully addressed before your manuscript can be considered further for publication in *eLife*. Essential revisions can be summarized as follows:

1) The study lacks a detailed analysis of axonal phenotypes in klc4 mutants and would be strengthened by analyzing single arbor dynamics and/or using additional morphological criteria to define branches and analyze axonal density.

2) The study lacks mechanistic insight into the roles of KLC4 and the phenotypes observed. Reviewers agree that at least some mechanistic experiments are necessary to explain how klc4 mutations cause axonal phenotypes. These include analyzing the transport of specific cargoes during axonal development and microtubule behavior during axonal branching. The localization and cell-autonomous function of klc4 could also be analyzed.

3) There is a disconnect between the description of sensory axonal defects in embryos and the characterization of behavioral defects in adults. Behavioral assays that rely on functional sensory neurons instead (touch-response in larvae for example) would be more appropriate for this study.

4) As described in detail by reviewer 2 (see below), multiple concerns about statistical analyses were raised and need to be addressed.

*Reviewer #1 (Recommendations for the authors):*

Overall, the manuscript is very well written and the conclusions are well supported by the data. The light sheet and swept confocal microscopy movies provide a unique view of sensory axon development in vivo and are beautiful! Below are a few questions / comments that I had while reading the manuscript:

1) The data in figures 3 and 5 show nicely that axon branches are not stabilized and that axonal growth is shifted posteriorly in klc4 mutants. Is the collapse of branches direction-dependent in the mutant? Is there a difference in branch stabilization/pruning depending on whether the branch elongates anteriorly or posteriorly?

2) The presence of apical protrusions crossing the midline in klc4 mutants suggests a defect in RB neuron polarity. Is the centrosome mislocalized in klc4 mutants?

3) The existence of both Klc4 isoforms is interesting, especially since corresponding transcripts seem to be differently regulated during development. Since the short isoform is not predicted to bind any cargo, is anything known about its functions? Are both isoforms expressed in RB neurons?

4) It would be interesting to analyze the localization of Klc4 in sensory neurons, although this might be difficult to do. Would it be possible to mosaically express low levels of tagged klc4 in mutant sensor neurons and analyze its distribution?

*Reviewer #2 (Recommendations for the authors):*

The authors use Crisper technology to generate a zebrafish line that lacks expression of KLC4. This animal is then used throughout, along with pharmacological inhibition of KLC binding to adaptor proteins, to address the roles of KLC4 in the development of the morphology of Rohon-Beard (RB) sensory neurons in embryos and assess behavioral alterations in adult fish. Modern imaging methods are applied to study neuronal morphology in embryos and a battery of behavioral tests in adults. The study provides first line of inquiry assessment of the role of KLC4 in the development of RB morphology, focusing on the failure to establish proper axon arbors in the periphery. While each data set has intrinsic value and is novel, none is followed up to provide mechanistic insights into KLC4 functions in neurons. The value of the submission rests in providing a characterization of the effects of KLC4 knockout, but it does not further the understanding of the mechanistic contributions of KLC4 in the processes impacted by the genetic deletion of KLC4. While the significance of the question posed is well presented and clear, multiple issues regarding interpretation, presentation and analysis decrease enthusiasm for the submission in its current form.

For the authors' consideration:

1. The current in situ data uses a riboprobe that does not differentiate between the short and long forms of the KLC4 transcript. Given the large difference between short and long transcripts of KLC4, and expected differences in function of the resultant proteins, it would assist in the interpretation of the results to assess short and long transcript expression in RB neurons. Ideally, this would be followed up by specific knock down of the long form to dissect its contributions from the short form.

2. At first evaluation, it is not clear how the axon branching metric (presented in Supplemental Figure 2) is specific for axon branching. In the images provided there seem to be consistently fewer axons present as a function of distance from the central fasicle in the KLC mutant and kinesore treated embryos (e.g., Figure 2A,B,D,E,H,I; Figure 4A,B). How can the metric dissociate the number of main axons that arrived at the ROI from branches made from those axons?

Based on the high quality of the imaging, it seems that it would be possible to use morphological criteria to define axon branches and measure the density along individual axons. This form of assessment would be much more stringent than the ROI crossing method used. Indeed, this form of assessment is used in the live imaging studies addressing rates or branch formation and stability.

3. Considering the above comment (2) regarding the consistent observable decrease in the apparent number of axons and their lengths into the periphery noted above, it is surprising that an increase in the rate of axon extension was detected in the live imaging analysis (Figure 3E). It is suggested that the authors analyze the density of axons and their total length as a function of distance from the central axon fascicle, noting that the ROI based approach in supplemental Figure 2 should serve this purpose by simply changing the distance from the fascicle into the periphery at constant intervals (currently only 65 microns was addressed).

Based on the images presented I would expect that a decrease in axon density and length will be determined in the KLC4 mutant. If so, then these data would stand in conflict with the observed increase rate of axon extension, but may indicate that earlier aspects of axon formation were also impacted in the KLC4 mutant. The formation of axons from the cell body may share mechanistic similarity to the emergence of branches, and thus may be delayed in the mutant due to failure to stabilize the new axon at its early stages of formation which could then result in the difference suggested by the images provided without being confounded by the subsequent increase in axon extension rate. It would also be insightful to perform this analysis at multiple points in ontogenesis.

4. An interpretational caveat is that with both the genetic mutant and the drug treatment all cells are expected to be affected. From the in situ is it not clear to this reviewer if there is no expression of KLC4 in the environment the axons/branches are migrating through. Might this issue be resolved using zebrafish genetics with lines allowing for specific knockdown in RB neurons? or some other approach to address whether the effects are cell intrinsic.

5. Statistical analysis. There are multiple concerns regarding the statistical presentation and analysis.

The section in the methods requires much elaboration, starting with which test was used to determine the normalcy of data sets, and addressed by subsequent comments.

Figures4 and 10 require statistical analysis of the data. "Plots using 95% confidence intervals (CI) were considered to be significant when the 95% CI ranges did not overlap." Is not considered a stringent analytical method, although a valid "rule of thumb". The authors are urged to seek assistance from a statistician.

6E, no 95% CI shown as claimed in the legend. The mean is not likely to be significant as the WT only have 3 points and one is an outlier relative to the other 2. I also do not see how the mean in the +/+ could be as shown, as claimed in the legend, it seems to be the median. I would estimate the mean to be somewhere around 20 um. For the mutant data, the distribution does not strike me as normal. Were these data analyzed for normality? Also, there is no statistical analysis shown, appreciating that with an n=3 for the WT this is problematic and power would have to be increased through greater sampling.

The legend to Figure 8 states that t-test with Mann-Whitney correction are used. I am not familiar with this correction for a t-test and was not able to find references to such a correction for t-test. Was Welch correction intended? The statistical description in the method is very sparse as presented and details should be presented. How the basic assumptions of the rather limited t-test were met by data prior to selection of specific t-test should be clearly denoted.

Figure 9. In panel C, a t-test is used but (1) the data are bounded values, and (2) the distribution in WT is clearly non normal with many data points at the boundary (i.e., 100%). A t-test is not applicable as basic assumptions are broken, and do not appear to have been determined. E and H have no p values or test shown in legend and it is not clear why individual data points are not shown here as in other panels. F and G have no test denoted. The meaning of * and multiples thereof is only defined for one panel. Panel G has *** not defined at all.

6. Strictly speaking, the authors are inferring contact repulsion but do not use the imaging data sets to directly address it (i.e., through direct analysis of the consequences of contacts between the two sister branches after growth cone bifurcation). The observed changes in angles between sister branches could be due to a variety of other mechanisms altering the ability of sister branches to generate larger angles.

The self-contact examples in Figure 6 are not particularly compelling. Perhaps, presentation of all relevant frames in the timelapse with greater annotation and description would assist a reader.

In Figure 6B, in contrast to what the text claims, this reviewer does not see a clear growth cone bifurcation followed by fasiculation due to lack of self-repulsion. It appears a small back branch from the second panel from the left aligns with the main axon.

Also, the "appearing to collapse onto one another" (line 232) could be interpreted as a consequence of the prior analysis of branch stability.

7. Figure 8D,E. "Comet Frequency" in comets/um/min. It is not clear if this reflects new comets observed during the imaging or all observed comets/um/min.

The imaging sets would benefit from additional analysis to extract the duration of EB3 comets (i.e., summed period spent polymerizing per comet) and total distance from initiation to loss of the comet.

It is not clear why central axons are analyzed for only these data sets and not others. As the focus of the paper is on peripheral axons the suggested additional data sets for peripheral axons would suffice.

As a whole, I would suggest that the data in Figure 8 from EB3 imaging indicate increased polymerization of plus tips along axons in the mutant. This is not immediately reconcilable with decreases branch stability as one would presume that microtubules would be more prone to catastrophe and depolymerization in conjunction with decreased branch stability. The data are broadly consistent with the detected increase in axon extension rate. However, analysis of EB3 comets needs to be provided in branches (collateral and bifurcated) in order to be linked to the main observed phenotype of decreased branch stability.

8. While of general interest, the behavioral assays do not address RB neuron function or circuitry, which is embryonic. The fit of these data to the rest of the study is not clear. Could tail startle be assessed (Liu and Hale (2017) Curr Biol 27:697-704) for addressing RB/RB-related circuit function? Any RB function assays would provide a much better fit to the rest of the data presented.

Importantly, these behavioral deficits cannot be attributed to changes in circuit formation, branching, axon extension etc as the mutant is not neuron specific, the relevant circuitry has not been analyzed, and no aspect of CNS neurons or glia or anything else has been analyzed. The concern is compounded by the differences in overall growth of the fish.

9. One fundamental aspect of the study that is missing is consideration of KLC4 localization in the RB neurons. Whether it localizes to the RB axons remains unknown, or whether it ever targets to branches. One would presume so, but this needs to be assessed. It is hard to interpret results with any degree of mechanistic insight without knowledge of the localization of the molecule addressed.

10. Final considerations: Overall, this submission contains interesting and valuable data sets that are however preliminary in nature as none is followed up to achieve a deeper mechanistic understanding. This manuscript presents a characterization of phenotypes in KLC4 mutant zebrafish RB neurons but does not provide insights into how KLC4 is mechanistically contributing to the observed phenotypes, or whether it is acting cell intrinsically. Having said this, the discussion carefully considers the potential implications of the individual findings and notes relevant future directions.

*Reviewer #3 (Recommendations for the authors):*

Further characterization of axonal defects:

– While the population-based methods for characterizing axonal defects are elegant and informative, these defects could potentially be further appreciated with single cell resolution, What do the arbors of individual neurons look like in these mutants? Does the mutant affect all sensory neurons equally, or is a sub-population of neurons more severely affected? It should be possible to analyze single neurons with transient transgenesis in the zebrafish model. This could be particularly helpful since aberrant fasciculation may make it difficult to distinguish multiple fasciculated axon branches when the whole population of neurons is labeled. Identifying defects at a single neuron level could also make it possible to carry out rescue experiments to address the cell autonomy of axon arborization defects.

Explanation of axonal defects mechanistically:

– The impact of this paper would be most enhanced by characterizing the axonal trafficking of cellular cargos in klc4 mutants, particularly if defects in specific trafficked cargoes could plausibly explain the axonal defects. In the discussion of the paper, the authors note that they have previously found that endosomal trafficking regulates branching. Characterizing endosomal trafficking, as well as trafficking of mitochondria, lysosomes, golgi, etc, thus has the potential to make this study substantially more mechanistically satisfying.

– The changes in microtubule dynamics of klc4 mutants are interesting, but how trafficking defects affect these dynamics, and how defects in MT dynamics affect branching, is unclear. To address how trafficking affects MTs, the authors could attempt to identify MTOCs in axons. For example, since golgi outposts are thought to be MTOCs, is their trafficking into axons affected by klc4 mutation? To address how MT defects affect branching, the authors could characterize MT dynamics during branching-for example, do mutants display any defects in the frequency of MT invasion into nascent branches?

Behavior:

– As mentioned above, the analyses of adult behavioral defects, while interesting and rigorously carried out, do not complement the morphological analysis in larval sensory neurons. It would be exciting if axonal arborization defects in a specific neuron type could be linked to behavioral defects. Could the authors attempt to characterize touch-response behaviors in larval fish? This would be a particularly exciting if it were possible to do cell-autonomous rescue experiments, thus directly testing if defects in sensory neuron morphology affect behavior.

---

## [Author Response]

Essential revisions1) The study lacks a detailed analysis of axonal phenotypes in klc4 mutants and would be strengthened by analyzing single arbor dynamics and/or using additional morphological criteria to define branches and analyze axonal density.

To get more detailed information about axon density, we increased our image sampling by using a grid ROI instead of a single line ROI. This analysis approach provides information about axon density as a function of dorsal-ventral location as suggested by Reviewer 2, and also gives information about both growth cone bifurcation and interstitial branching. Details are described in response to Reviewer 2, points 2 and 3 below, and data added to Figure 2F, G, and Supplemental Figure 2. Although all neurons were labeled in our live imaging of axon growth and branching experiments, we were able to distinguish axons from individual neurons, and the analysis of branch dynamics was done on individual neurons. Further description of our reasoning on this point is in our response to Reviewer 3, point 1.

2) The study lacks mechanistic insight into the roles of KLC4 and the phenotypes observed. Reviewers agree that at least some mechanistic experiments are necessary to explain how klc4 mutations cause axonal phenotypes. These include analyzing the transport of specific cargoes during axonal development and microtubule behavior during axonal branching. The localization and cell-autonomous function of klc4 could also be analyzed.

We added new experiments to image endosomal dynamics (new Figure 8), trafficking of Rab5-labeled endosomes from the cell body to branch points (new Figure 8 —figure supplement 1A, 1B), microtubule behavior during axon branching (new Figure 10), and acetylation of microtubules in nascent branches (new Figure 10). We chose Rab5-labeled endosomes because of our previous work showing a correlation between their transport and RB axon branching. Although the *klc4* mutants did not have a decrease in endosome delivery to branch points, high-speed imaging of vesicle dynamics revealed some alterations in transport of these vesicles in *klc4* mutants. These alterations in endosome transport are not likely to be the only effects of *klc4* mutation on cargo dynamics or to fully explain the mechanism by which KLC promotes branch stabilization. However, an exhaustive analysis of all potential cargos would require many months to years of work and is outside the scope of this manuscript. Further description is in response to Reviewer 3, point 2.

We also added new experiments analyzing microtubules in nascent branches (new Figure 10). We used live imaging with EB3-GFP to measure frequency of comets entering nascent branches. Although comet frequency in established axons is increased in *klc4* mutants, there is no difference in frequency of comets entering nascent branches, suggesting a relative decrease. We used anti-acetylated tubulin labeling to show that *klc4* mutants have fewer nascent branches with stable, acetylated tubulin. Further description is in our response to Reviewer 2, point 7 and Reviewer 3, point 3.

As described in our response to Reviewer 2, point 4, we believe the effects of *klc4* mutation are likely cell autonomous because *klc4* is not expressed in the somites or the skin, which are the tissues in contact with RB peripheral axons, and RB neurons are the predominant cells in the dorsal spinal cord that express *klc4* at the stages when we see the axon phenotypes.

As described in response to Reviewer 1, point 4 and Reviewer 2, point 9, there are major obstacles that prevent visualization of KLC4 protein localization within neurons. There are no available antibodies to zebrafish KLC4. We tried using expression of KLC4-GFP fusion constructs (as suggested by Reviewer 1), but expression leads to diffuse labeling throughout the cytoplasm. This problem has been previously reported by others attempting to visualize kinesin-1 heavy chains (Yang et al. 2019, *Traffic* 20:851-866). Because the vast majority of kinesin light chains and heavy chains exist in an auto-inhibited conformation that is freely diffusible, the diffuse signal resulting from expression of tagged proteins obscures visualization of motors bound to cargos. Thus, we are unfortunately not able to image KLC4 protein localization with current available approaches.

3) There is a disconnect between the description of sensory axonal defects in embryos and the characterization of behavioral defects in adults. Behavioral assays that rely on functional sensory neurons instead (touch-response in larvae for example) would be more appropriate for this study.

We added new experiments testing the larval touch response. We found that *klc4* mutant larvae showed increased number of swim bouts and longer duration swim runs after a touch stimulus, suggesting the mutant larvae are hypersensitive to touch. We added these data to Figure 11. We moved the data on adult body size and feeding behavior to Figure 11 —figure supplement 1A-C. However, we kept the data on adult anxiety-like behavior in the main Figures 11 and 12. We believe these results will be of interest to the community and potentially useful as a new model to study stress response behaviors and circuits. Further explanation of our reasoning for keeping these data in the paper is in the response to Reviewer 2, point 8 and Reviewer 3, point 4.

4) As described in detail by reviewer 2 (see below), multiple concerns about statistical analyses were raised and need to be addressed.

We address each concern in the response to Reviewer 2, point 5.

Reviewer #1 (Recommendations for the authors):Overall, the manuscript is very well written and the conclusions are well supported by the data. The light sheet and swept confocal microscopy movies provide a unique view of sensory axon development in vivo and are beautiful! Below are a few questions / comments that I had while reading the manuscript:1) The data in figures 3 and 5 show nicely that axon branches are not stabilized and that axonal growth is shifted posteriorly in klc4 mutants. Is the collapse of branches direction-dependent in the mutant? Is there a difference in branch stabilization/pruning depending on whether the branch elongates anteriorly or posteriorly?

We scored the directionality of the axon branches that failed to stabilize and did not find a significant difference in the percent that were anteriorly or posteriorly directed. We have added this result to the text of the Results.

2) The presence of apical protrusions crossing the midline in klc4 mutants suggests a defect in RB neuron polarity. Is the centrosome mislocalized in klc4 mutants?

This is an interesting question. We previously showed a correlation between mislocalized centrosomes and apical protrusions with two other manipulations that lead to apical protrusions in RB neurons: inhibition of LIM homeodomain transcription factors and lamininA1 mutation (Andersen and Halloran, 2012, *Development* 139:3590-3599). We expect that we would also see apical mislocalization of the centrosome in *klc4* mutants. However, because showing a third condition under which these two events are correlated would be a relatively incremental conceptual advance, and because centrosome imaging is one of several time-intensive experiments requested by reviewers, we focused our efforts on the experiments deemed to be essential revisions (imaging of cargo transport and microtubule dynamics during branching, and larval behavior experiments).

3) The existence of both Klc4 isoforms is interesting, especially since corresponding transcripts seem to be differently regulated during development. Since the short isoform is not predicted to bind any cargo, is anything known about its functions? Are both isoforms expressed in RB neurons?

We made a riboprobe that recognizes sequence found only in the long isoform. We found that the expression pattern of the long isoform is very similar to that of the riboprobe that binds both isoforms, indicating that the long isoform is likely present in all *klc4*-expressing neurons, including RB neurons. We added those data in Figure 1. The short isoform does not have unique sequence that isn’t present in the long isoform, so we are not able to make a short isoform specific riboprobe. To our knowledge, nothing is known about the function of the short isoform.

4) It would be interesting to analyze the localization of Klc4 in sensory neurons, although this might be difficult to do. Would it be possible to mosaically express low levels of tagged klc4 in mutant sensor neurons and analyze its distribution?

We tried using expression of KLC4-GFP fusion constructs, but expression leads to diffuse labeling throughout the cytoplasm. This problem has been previously reported by others attempting to visualize kinesin-1 heavy chains (Yang et al. 2019, *Traffic* 20:851-866). Because the vast majority of kinesin light chains and heavy chains exist in an auto-inhibited conformation that is freely diffusible, the diffuse signal resulting from expression of tagged proteins obscures visualization of motors bound to cargos. Thus, we are unfortunately not able to image KLC4 protein localization with current available approaches.

Reviewer #2 (Recommendations for the authors):The authors use Crisper technology to generate a zebrafish line that lacks expression of KLC4. This animal is then used throughout, along with pharmacological inhibition of KLC binding to adaptor proteins, to address the roles of KLC4 in the development of the morphology of Rohon-Beard (RB) sensory neurons in embryos and assess behavioral alterations in adult fish. Modern imaging methods are applied to study neuronal morphology in embryos and a battery of behavioral tests in adults. The study provides first line of inquiry assessment of the role of KLC4 in the development of RB morphology, focusing on the failure to establish proper axon arbors in the periphery. While each data set has intrinsic value and is novel, none is followed up to provide mechanistic insights into KLC4 functions in neurons. The value of the submission rests in providing a characterization of the effects of KLC4 knockout, but it does not further the understanding of the mechanistic contributions of KLC4 in the processes impacted by the genetic deletion of KLC4. While the significance of the question posed is well presented and clear, multiple issues regarding interpretation, presentation and analysis decrease enthusiasm for the submission in its current form.For the authors' consideration:1. The current in situ data uses a riboprobe that does not differentiate between the short and long forms of the KLC4 transcript. Given the large difference between short and long transcripts of KLC4, and expected differences in function of the resultant proteins, it would assist in the interpretation of the results to assess short and long transcript expression in RB neurons. Ideally, this would be followed up by specific knock down of the long form to dissect its contributions from the short form.

We made a riboprobe that recognizes sequence found only in the long isoform. We found that the expression pattern of the long isoform is very similar to that of the riboprobe that binds both isoforms, indicating that the long isoform is likely present in all *klc4*-expressing neurons, including RB neurons. We added those data in Figure 1. The short isoform does not have unique sequence that isn’t present in the long isoform, so we are not able to make a short isoform specific riboprobe. We also do not have a means to specifically knock down the short isoform. Splice blocking morpholinos could potentially be used to knock down the long isoform, but given the issues with morpholinos and interpretation of morpholino phenotypes, this would not be a definitive experiment.

2. At first evaluation, it is not clear how the axon branching metric (presented in Supplemental Figure 2) is specific for axon branching. In the images provided there seem to be consistently fewer axons present as a function of distance from the central fasicle in the KLC mutant and kinesore treated embryos (e.g., Figure 2A,B,D,E,H,I; Figure 4A,B). How can the metric dissociate the number of main axons that arrived at the ROI from branches made from those axons?

The metric does not distinguish between primary peripheral axons and those arising from secondary, tertiary or higher order branches. Most RB peripheral axons form as a branch from the central axon close to the cell body, and often this primary branch bifurcates shortly after spinal cord exit. Thus, all the axon segments in the periphery should be considered branches. We modified our language in the manuscript to avoid use of the term “main axon” because there is no clear definition of this term.

Based on the high quality of the imaging, it seems that it would be possible to use morphological criteria to define axon branches and measure the density along individual axons. This form of assessment would be much more stringent than the ROI crossing method used. Indeed, this form of assessment is used in the live imaging studies addressing rates or branch formation and stability.

To get more detailed information about axon density, we increased our image sampling by using a grid ROI instead of a single line ROI. This analysis approach provides information about axon density as a function of dorsal-ventral location (as suggested by the reviewer in point 3 below). This approach also provides information about both growth cone bifurcation and interstitial branching. We added the new data to Figure 2F, G, and Figure 2 —figure supplement 1.

3. Considering the above comment (2) regarding the consistent observable decrease in the apparent number of axons and their lengths into the periphery noted above, it is surprising that an increase in the rate of axon extension was detected in the live imaging analysis (Figure 3E). It is suggested that the authors analyze the density of axons and their total length as a function of distance from the central axon fascicle, noting that the ROI based approach in supplemental Figure 2 should serve this purpose by simply changing the distance from the fascicle into the periphery at constant intervals (currently only 65 microns was addressed).

This is an excellent suggestion, and we now added this additional, multiple ROI analysis.

Based on the images presented I would expect that a decrease in axon density and length will be determined in the KLC4 mutant. If so, then these data would stand in conflict with the observed increase rate of axon extension, but may indicate that earlier aspects of axon formation were also impacted in the KLC4 mutant. The formation of axons from the cell body may share mechanistic similarity to the emergence of branches, and thus may be delayed in the mutant due to failure to stabilize the new axon at its early stages of formation which could then result in the difference suggested by the images provided without being confounded by the subsequent increase in axon extension rate. It would also be insightful to perform this analysis at multiple points in ontogenesis.

There are fewer axons present in the mutants because there is less branching. In both wild type and mutants, axons that are shorter at one snapshot in time likely formed more recently than the longer axons. Live imaging is required to make conclusions about extension rate, and our live imaging showed that in the mutants, the axons that do form grow faster than wild type axons. We previously showed that the peripheral axons most often arise as a branch off the central axon (Andersen et al. 2011, *Neural Dev* 6:27), and we do believe that process shares mechanistic similarity to the formation of subsequent branches. Interestingly, the grid ROI analysis showed fewer branches in the mutants at all dorsal-ventral levels, indicating that the initial formation of peripheral axons may also be affected in the mutants, as the reviewer suggests. We added discussion of our interpretation of these results to the manuscript.

4. An interpretational caveat is that with both the genetic mutant and the drug treatment all cells are expected to be affected. From the in situ is it not clear to this reviewer if there is no expression of KLC4 in the environment the axons/branches are migrating through. Might this issue be resolved using zebrafish genetics with lines allowing for specific knockdown in RB neurons? or some other approach to address whether the effects are cell intrinsic.

We believe the effects in the mutant are likely cell autonomous. While it may be difficult to see from images captured at one focal plane, we can clearly distinguish the different tissues when looking through the microscope. We see no *klc4* expression in the somites or the skin cells, which are the tissues that RB peripheral axons contact while growing. In addition, the RB neurons are the predominant cell type in the dorsal spinal cord expressing *klc4* at the stages we see the axon phenotypes.

5. Statistical analysis. There are multiple concerns regarding the statistical presentation and analysis.The section in the methods requires much elaboration, starting with which test was used to determine the normalcy of data sets, and addressed by subsequent comments.

On every analyzed dataset tests of normality were run before a method to test significant differences was decided. We used Prism’s suite of normality tests, which include Anderson-Darling test, D’Agostino and Pearson test, and Shapiro-Wilk test. In nearly all cases these tests were in agreement about whether the data were normal or not normal. Occasionally, one test of the three would suggest data were non-normal when the other two tests would count them as normal. In these cases, the data were always considered to be non-normal. We have elaborated this process in the revised Methods section.

Figures4 and 10 require statistical analysis of the data. "Plots using 95% confidence intervals (CI) were considered to be significant when the 95% CI ranges did not overlap." Is not considered a stringent analytical method, although a valid "rule of thumb". The authors are urged to seek assistance from a statistician.

We consulted with a statistician, who advised us to calculate the area under the curve. For Figure 4, in the portion of the graph where the 95% confidence intervals do not overlap (posterior angles -20 to -45, area shaded in Figure 4D), we calculated the area under the curve, which gives us a mean and standard error of the mean and allows a t-test to be run on the data. The t-test showed that the difference in area under the curve was significant (p=0.0047). In Figure 12 (formerly Figure 10), we calculated the area under the curve in panels B-F and found all to be significant, with p values p=0.0005 or p<0.0001. We added these analyses to the Methods section and figure legends.

6E, no 95% CI shown as claimed in the legend. The mean is not likely to be significant as the WT only have 3 points and one is an outlier relative to the other 2. I also do not see how the mean in the +/+ could be as shown, as claimed in the legend, it seems to be the median. I would estimate the mean to be somewhere around 20 um. For the mutant data, the distribution does not strike me as normal. Were these data analyzed for normality? Also, there is no statistical analysis shown, appreciating that with an n=3 for the WT this is problematic and power would have to be increased through greater sampling.

We regret the inaccuracy of the legend which likely occurred during changes to that figure. The figure has been updated to show the mean with standard error of the mean. As the reviewer noted, these data were not analyzed for significance because of the paucity of fasciculation events in wildtype (all events are shown). We did not report the normality test because we were not running a statistical test on the data. However, a normality test showed that mutant distribution was normal by Anderson-Darling, D’Agostino and Pearson, and Shapiro-Wilk. Normality could not be assessed for wild type because of the small number of fasciculation events. Our statistical consultant confirmed that it would not be appropriate to run a statistical test on the data because the events are so rare in wild type. They advised us to note in the legend that length measurements are conditional on event occurring, which we have done. We believe that reporting these measurements is still informative to readers even if they can’t be tested for statistical significance.

The legend to Figure 8 states that t-test with Mann-Whitney correction are used. I am not familiar with this correction for a t-test and was not able to find references to such a correction for t-test. Was Welch correction intended? The statistical description in the method is very sparse as presented and details should be presented. How the basic assumptions of the rather limited t-test were met by data prior to selection of specific t-test should be clearly denoted.

We apologize for the inaccurate description. The Mann Whitney test is not a corrected t-test, but rather an alternative to the t-test for non-normal data. This test is the same as the Wilcoxon rank sum test. This does not change the results, as the Mann-Whitney test is still an appropriate test for these data. We added more description to the statistical section of the Methods. Before selection of a test, the data were tested for normality by the previously described methods. In most cases, data were not normally distributed in all samples, leading to use of the Mann-Whitney test. In cases where data were determined to be normal, a standard t-test was used. Data tested by Mann-Whitney or t-tests were independent of each other and comprised ordinal or continuous numbers. Data tested by t-test were confirmed to have no difference in variance by performing an F test to compare variances.

Figure 9. In panel C, a t-test is used but (1) the data are bounded values, and (2) the distribution in WT is clearly non normal with many data points at the boundary (i.e., 100%). A t-test is not applicable as basic assumptions are broken, and do not appear to have been determined. E and H have no p values or test shown in legend and it is not clear why individual data points are not shown here as in other panels. F and G have no test denoted. The meaning of * and multiples thereof is only defined for one panel. Panel G has *** not defined at all.

We regret that the legend for Figure 9, Panel C (now Figure 11 —figure supplement 1C) was incorrect as a Mann-Whitney test was performed and not a t-test. The Mann-Whitney test accepts ordinal values, which should include bounded variables. The Mann-Whitney test is also equivalent to the Wilcoxon rank sum test. To be thorough, we also performed a Kruskal-Wallis test, which also works on independent data that are non-normal and ordinal or continuous. We found that the Kruskal-Wallis test agreed with the Mann-Whitney test and found that there was a significant difference in the number of larvae eaten between wildtype and mutant fish, with p<0.0001.

For E and H (now Figure 11 E, H), these graphs represent 3 individual experiments. Multiple animals were tested in each trial, but the N is the number of individual behavioral trial days. N=3 is too low to test for normality, therefore it was not appropriate to perform a statistical analysis on the data. The Mann-Whitney test (used for non-normal data) fails with an N this low—if the N is less than 7, the Mann-Whitney will always give a p value greater than 0.05 no matter how much the groups differ. We were able to test significance of the total time spent in the upper zone (Figure 11G) and the time spent freezing (Figure 11I). We believe it is informative to also report the percent fish that entered the upper zone in each experiment (Figure 11E) and the percent that froze (Figure 11H), even though these can’t be tested for significance. We have updated the data display to show individual points for each experiment, along with mean and standard error of the mean. We have updated the legends to include all statistical test descriptions and p values.

6. Strictly speaking, the authors are inferring contact repulsion but do not use the imaging data sets to directly address it (i.e., through direct analysis of the consequences of contacts between the two sister branches after growth cone bifurcation). The observed changes in angles between sister branches could be due to a variety of other mechanisms altering the ability of sister branches to generate larger angles.

The reviewer’s point is well taken. We have modified our language in the text accordingly.

The self-contact examples in Figure 6 are not particularly compelling. Perhaps, presentation of all relevant frames in the timelapse with greater annotation and description would assist a reader.

We updated the images in Figure 6 to show a clearer example of self-contact. Space constraints prohibit including all relevant frames in the timelapse, but they can be viewed in the accompanying video.

In Figure 6B, in contrast to what the text claims, this reviewer does not see a clear growth cone bifurcation followed by fasiculation due to lack of self-repulsion. It appears a small back branch from the second panel from the left aligns with the main axon.

We have updated the images in Figure 6B to more clearly show branches collapsing onto each other.

Also, the "appearing to collapse onto one another" (line 232) could be interpreted as a consequence of the prior analysis of branch stability.

When following the dynamic axon behaviors in the movies, we can distinguish between branches that have retracted versus those that fasciculate onto other axons.

7. Figure 8D,E. "Comet Frequency" in comets/um/min. It is not clear if this reflects new comets observed during the imaging or all observed comets/um/min.

These data include all observed comets. We clarified this point in the figure legend (now Figure 9).

The imaging sets would benefit from additional analysis to extract the duration of EB3 comets (i.e., summed period spent polymerizing per comet) and total distance from initiation to loss of the comet.

We now include data on both distance (in microns) and duration (in seconds) of comet runs in Figure 9 —figure supplement 1C-F, and describe the results and interpretation in the Results section.

It is not clear why central axons are analyzed for only these data sets and not others. As the focus of the paper is on peripheral axons the suggested additional data sets for peripheral axons would suffice.

Because the focus is on peripheral axons, we moved the data on EB3-GFP comets in central axons to Figure 9 —figure supplement 1.

As a whole, I would suggest that the data in Figure 8 from EB3 imaging indicate increased polymerization of plus tips along axons in the mutant. This is not immediately reconcilable with decreases branch stability as one would presume that microtubules would be more prone to catastrophe and depolymerization in conjunction with decreased branch stability. The data are broadly consistent with the detected increase in axon extension rate. However, analysis of EB3 comets needs to be provided in branches (collateral and bifurcated) in order to be linked to the main observed phenotype of decreased branch stability.

We used two approaches to analyze microtubules in nascent branches (new Figure 10). First, we used EB3-GFP to image microtubule polymerization into nascent branches. Although comet frequency in established axons is increased in *klc4* mutants, there is no difference in frequency of comets entering nascent branches, suggesting a relative decrease. Second, we used anti-acetylated tubulin labeling to show that nascent branches in *klc4* mutants are less likely to have stable, acetylated tubulin. We added further interpretation of these data to the Discussion section.

8. While of general interest, the behavioral assays do not address RB neuron function or circuitry, which is embryonic. The fit of these data to the rest of the study is not clear. Could tail startle be assessed (Liu and Hale (2017) Curr Biol 27:697-704) for addressing RB/RB-related circuit function? Any RB function assays would provide a much better fit to the rest of the data presented.Importantly, these behavioral deficits cannot be attributed to changes in circuit formation, branching, axon extension etc as the mutant is not neuron specific, the relevant circuitry has not been analyzed, and no aspect of CNS neurons or glia or anything else has been analyzed. The concern is compounded by the differences in overall growth of the fish.

We added new experiments testing the larval touch response at 3 dpf, a behavior that is mediated by sensory neurons. We found that *klc4* mutant larvae showed increased number of swim bouts and longer duration swim runs after a touch stimulus, suggesting the mutant larvae are hypersensitive to touch. We added these data to Figure 10. We moved the data on adult body size and feeding behavior to Supplemental Figure 10. However, we kept the data on adult anxiety-like behavior in the main Figures 10 and 11. It is true that these behaviors cannot be linked directly to cell biological defects in RB neurons. We are not claiming a direct link. Instead we are making use of two separate approaches to address questions about KLC4 function: (1) We take advantage of the superficial position of the RB neurons, which makes them ideal for imaging, and their distinctly polarized morphology to uncover cell biological processes influenced by KLC4. (2) We use adult behavioral analysis to more broadly assess roles of KLC4 in other neural circuits. We believe that publication of the adult behavior phenotypes will benefit the scientific community and be useful as a new model to study stress response behaviors and circuits.

9. One fundamental aspect of the study that is missing is consideration of KLC4 localization in the RB neurons. Whether it localizes to the RB axons remains unknown, or whether it ever targets to branches. One would presume so, but this needs to be assessed. It is hard to interpret results with any degree of mechanistic insight without knowledge of the localization of the molecule addressed.

As mentioned in response to reviewer 1, we tried using expression of KLC4-GFP fusion constructs, but expression leads to diffuse labeling throughout the cytoplasm. This problem has been previously reported by others attempting to visualize kinesin-1 heavy chains (Yang et al. 2019, *Traffic* 20:851-866). Because the vast majority of kinesin light chains and heavy chains exist in an auto-inhibited conformation that is freely diffusible, the diffuse signal resulting from expression of tagged proteins obscures visualization of motors bound to cargos. Thus, we are unfortunately not able to image KLC4 protein localization with current available approaches.

10. Final considerations: Overall, this submission contains interesting and valuable data sets that are however preliminary in nature as none is followed up to achieve a deeper mechanistic understanding. This manuscript presents a characterization of phenotypes in KLC4 mutant zebrafish RB neurons but does not provide insights into how KLC4 is mechanistically contributing to the observed phenotypes, or whether it is acting cell intrinsically. Having said this, the discussion carefully considers the potential implications of the individual findings and notes relevant future directions.Reviewer #3 (Recommendations for the authors):Further characterization of axonal defects:– While the population-based methods for characterizing axonal defects are elegant and informative, these defects could potentially be further appreciated with single cell resolution, What do the arbors of individual neurons look like in these mutants? Does the mutant affect all sensory neurons equally, or is a sub-population of neurons more severely affected? It should be possible to analyze single neurons with transient transgenesis in the zebrafish model. This could be particularly helpful since aberrant fasciculation may make it difficult to distinguish multiple fasciculated axon branches when the whole population of neurons is labeled. Identifying defects at a single neuron level could also make it possible to carry out rescue experiments to address the cell autonomy of axon arborization defects.

We did analyze individual neurons labeled with transient transgenesis for the Rab5 and microtubule imaging experiments, and the single-cell labeling in Figure 7 to follow the trajectory of apical processes. As the reviewer points out, this is important because fasciculation of multiple axons would confound results. We used stable transgenics for the live imaging of axon growth, fasciculation and branch dynamics. This was absolutely necessary for the fasciculation analysis because all axons must be labeled to detect non-self-fasciculations. We are able to do the growth rate and branch dynamics analyses on individual neurons using the stable transgenics because we can easily resolve individual neurons by following videos of axons growing over time.

Explanation of axonal defects mechanistically:– The impact of this paper would be most enhanced by characterizing the axonal trafficking of cellular cargos in klc4 mutants, particularly if defects in specific trafficked cargoes could plausibly explain the axonal defects. In the discussion of the paper, the authors note that they have previously found that endosomal trafficking regulates branching. Characterizing endosomal trafficking, as well as trafficking of mitochondria, lysosomes, golgi, etc, thus has the potential to make this study substantially more mechanistically satisfying.

We added new experiments to image endosomal dynamics (new Figure 8), trafficking of Rab5-labeled endosomes from the cell body to branch points (new Figure 8 —figure supplement 1), microtubule behavior during axon branching (new Figure 10), and acetylation of microtubules in nascent branches (new Figure 10). We chose Rab5-labeled endosomes because of our previous work showing a correlation between their transport and RB axon branching. Although the *klc4* mutants did not have a decrease in endosome delivery to branch points, high-speed imaging of vesicle dynamics revealed some alterations in transport of these vesicles in *klc4* mutants. These alterations in endosome transport are not likely to be the only effects of *klc4* mutation on cargo dynamics or to fully explain the mechanism by which KLC promotes branch stabilization. However, an exhaustive analysis of all potential cargos would require many months to years of work and is outside the scope of this manuscript.

– The changes in microtubule dynamics of klc4 mutants are interesting, but how trafficking defects affect these dynamics, and how defects in MT dynamics affect branching, is unclear. To address how trafficking affects MTs, the authors could attempt to identify MTOCs in axons. For example, since golgi outposts are thought to be MTOCs, is their trafficking into axons affected by klc4 mutation? To address how MT defects affect branching, the authors could characterize MT dynamics during branching-for example, do mutants display any defects in the frequency of MT invasion into nascent branches?

We have tried to detect MTOCs using an antibody to γ tubulin, but that labeling in whole embryos has a very low signal to noise. It was not possible for us to detect signal above noise. We added new experiments analyzing microtubules in nascent branches (new Figure 10). We used live imaging with EB3-GFP to measure frequency of comets entering nascent branches. Although comet frequency in established axons is increased in *klc4* mutants, there is no difference in frequency of comets entering nascent branches, suggesting a relative decrease. We used anti-acetylated tubulin labeling to show that *klc4* mutants have fewer nascent branches with stable, acetylated tubulin.

Behavior:– As mentioned above, the analyses of adult behavioral defects, while interesting and rigorously carried out, do not complement the morphological analysis in larval sensory neurons. It would be exciting if axonal arborization defects in a specific neuron type could be linked to behavioral defects. Could the authors attempt to characterize touch-response behaviors in larval fish? This would be a particularly exciting if it were possible to do cell-autonomous rescue experiments, thus directly testing if defects in sensory neuron morphology affect behavior.

This is an excellent suggestion and we have now added larval touch response assays (Figure 11). We found that *klc4* mutant larvae showed increased number of swim bouts and longer duration swim runs after a touch stimulus, suggesting the mutant larvae are hypersensitive to touch. We added these data to Figure 11. We moved the data on adult body size and feeding behavior to Figure 11 —figure supplement 1. However, we kept the data on adult anxiety-like behavior in the main Figures 11 and 12. We believe these results will be of interest to the community and potentially useful as a new model to study stress response behaviors and circuits.